# MULTI-TASK NEURAL PROCESSES

## ABSTRACT

Neural processes have recently emerged as a class of powerful neural latent variable models that combine the strengths of neural networks and stochastic processes. As they can encode contextual data in the network's function space, they offer a new way to model task relatedness in multi-task learning. To study its potential, we develop *multi-task neural processes*, a new variant of neural processes for multi-task learning. In particular, we propose to explore transferable knowledge from related tasks in the function space to provide inductive bias for improving each individual task. To do so, we derive the function priors in a hierarchical Bayesian inference framework, which enables each task to incorporate the shared knowledge provided by related tasks into its context of the prediction function. Our multi-task neural processes methodologically expand the scope of vanilla neural processes and provide a new way of exploring task relatedness in function spaces for multi-task learning. The proposed multi-task neural processes are capable of learning multiple tasks with limited labeled data and in the presence of domain shift. We perform extensive experimental evaluations on several benchmarks for the multi-task regression and classification tasks. The results demonstrate the effectiveness of multi-task neural processes in transferring useful knowledge among tasks for multi-task learning and superior performance in multi-task classification and brain image segmentation[1].

## 1 INTRODUCTION

As deep neural networks are black-box function approximations, it is difficult to introduce prior domain or expert knowledge into a prediction function (Jakkala, 2021). In contrast, Gaussian processes (Rasmussen, 2003) explicitly define distributions over functions and perform inference over these functions given some training examples. This enables reliable and flexible decision-making. However, Gaussian processes can suffer from high computational complexity due to the manipulation of kernel matrices. Therefore, there has been continuous interest in bringing together neural networks and Gaussian processes (Damianou & Lawrence, 2013; Wilson et al., 2016; Garnelo et al., 2018a; Jakkala, 2021) into so-called neural processes.

Neural processes (Garnelo et al., 2018b) combine the computational efficiency of neural networks with the uncertainty quantification of stochastic processes. They are a class of neural latent variables model, which deploy a deep neural network to encode context observations into a latent stochastic variable to model prediction functions. Neural processes provide an elegant formalism to efficiently and effectively incorporate multiple datasets into learning distributions over functions. This formalism is also promising in multi-task learning to improve individual tasks by transferring useful contextual knowledge among related tasks. Their capability of estimating uncertainty over predictions also makes them well-suited for multi-task learning with limited data, where each task has only a few training samples. However, neural processes rely on the implicit assumption that the context and target sets are from the same distribution and can be aggregated by a simple average pooling operation (Kim et al., 2019; Volpp et al., 2020). This makes it non-trivial to directly apply neural processes to modeling multiple heterogeneous tasks from different domains, where the context data of different tasks are from distinctive distributions (Long et al., 2017).

---

[1] Our code is available in https://anonymous.4open.science/r/Multi-Task-Neural-Processes/.

In this paper, we develop *multi-task neural processes* (MTNPs), a methodological extension of neural processes for multi-task learning, which fills the theoretical gap of neural processes for multi-task learning. Particularly, we propose to explore task relatedness in the function space by specifying the function priors in a hierarchical Bayesian inference framework. The shared knowledge from related tasks is incorporated into the context of each individual task, which serves as the inductive bias for making predictions in this task. The hierarchical architecture allows us to design expressive data-dependent priors. This enables the model to capture the complex task relationships in multi-task learning. By leveraging hierarchical modeling, multi-task neural processes are capable of exploring shared knowledge among related tasks in a principled way by specifying the function prior.

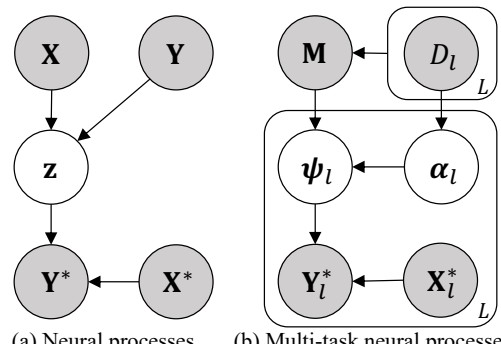

(a) Neural processes    (b) Multi-task neural processes

Figure 1: Graphical illustration of neural processes and multi-task neural processes. Shaded nodes indicate observed variables, and white nodes indicate the introduced latent variables.

We validate the effectiveness of the proposed multi-task neural processes by extensive experiments in both multi-task classification and regression. The results demonstrate that multi-task neural processes can effectively capture task relatedness in the function space and consistently improve the performance of each individual task, especially in the limited data regime.

## 2 PRELIMINARIES: NEURAL PROCESSES

In this section, we briefly review vanilla neural processes (Garnelo et al., 2018b) based on which we derive our multi-task neural processes.

Let a data set be given composed of training samples and corresponding labels. In order to better reflect the desired model behaviour at test time (Garnelo et al., 2018b), the training data is split into the context set $D = (\mathbf{X}, \mathbf{Y})$ and a target set $D^* = (\mathbf{X}^*, \mathbf{Y}^*)$, where $\mathbf{X} = \{\mathbf{x}_1, \cdots, \mathbf{x}_n\}$ is a subset of the training data, $\mathbf{Y} = \{\mathbf{y}_1, \cdots, \mathbf{y}_n\}$ the corresponding set of labels and $n$ the size of the context set. To ensure both sets have the same data distribution, the context set is split from the target set. Now, given the context set $(\mathbf{X}, \mathbf{Y})$, we would like to estimate a function that can make predict the labels $\mathbf{Y}^*$ for target samples $\mathbf{X}^*$.

In general, we define a stochastic process by a random function $f : \mathcal{X} \to \mathcal{Y}$. Given the context set, we define the joint distribution over the function values $\{f(\mathbf{x}_1), \cdots, f(\mathbf{x}_n)\}$, which in Gaussian processes is a multivariate Gaussian distribution parameterized by a kernel function. The rationale of neural processes is to extract knowledge from the context set to specify the prior over the prediction function. Instead of a kernel function, neural processes adopt a deep neural network to define the prior distribution. Specifically, the model introduces a latent variable $\mathbf{z}$ to account for uncertainty in the predictions of $\mathbf{Y}^*$. The observed context set is encoded into the latent variable which is conditioned on the context set, i.e. it follows the prior distribution $p(\mathbf{z}|\mathbf{X}, \mathbf{Y})$. The latent variable $\mathbf{z}$ is a high-dimensional random vector parameterising the stochastic process by $f(\mathbf{X}^*) = g(\mathbf{X}^*, \mathbf{z})$. The function $g(\cdot, \cdot)$ is an extra fixed and learnable decoder function, which is also implemented by a neural network. Thus, the neural process model can be formulated as follows:

$$p(\mathbf{Y}^*|\mathbf{X}^*, \mathbf{X}, \mathbf{Y}) = \int p(\mathbf{Y}^*|\mathbf{X}^*, \mathbf{z})p(\mathbf{z}|\mathbf{X}, \mathbf{Y})d\mathbf{z}. \tag{1}$$

The graphical model for neural processes is shown in Figure 1 (a).

The neural process model is optimized using amortized variational inference. Let $q(\mathbf{z}|\mathbf{X}^*, \mathbf{Y}^*)$ be a variational posterior of the latent variable $\mathbf{z}$. The evidence lower-bound (ELBO) for neural processes is given as follows:

$$\log p(\mathbf{Y}^*|\mathbf{X}^*, \mathbf{X}, \mathbf{Y}) \geq \mathbb{E}_{q(\mathbf{z}|\mathbf{X}^*, \mathbf{Y}^*)}[p(\mathbf{Y}^*|\mathbf{X}^*, \mathbf{z})] - \mathbb{D}_{\mathrm{KL}}[q(\mathbf{z}|\mathbf{X}^*, \mathbf{Y}^*)||p(\mathbf{z}|\mathbf{X}, \mathbf{Y})]. \tag{2}$$

In neural processes, the function space defined by deep neural networks allows the model to extract deep features while retaining a probabilistic interpretation (Jakkala, 2021). In multi-task learn-

ing, usually different tasks can be from different domains and have their specific data distributions (Lawrence & Platt, 2004; Long et al., 2017). Due to the complex data structure of multi-task learning, it is not straightforward to explore task relatedness in such function spaces. In this paper, we aim to extend the methodology of the neural process to the scenarios of multi-task learning to learn the shared knowledge among tasks for improving individual tasks.

## 3 MULTI-TASK NEURAL PROCESSES

The common setup of multi-task learning is that there are multiple related tasks for which we would like to improve the learning of each individual one by sharing information across the different tasks (Williams et al., 2007). Multi-task learning has been studied under different settings (Requeima et al., 2019; Williams et al., 2007; Lawrence & Platt, 2004; Yu et al., 2005; Long et al., 2017). In this paper, we tackle the multi-input multi-output setting, where each task has a different distribution while different tasks share the same target space (Lawrence & Platt, 2004; Yu et al., 2005; Long et al., 2017; Zhang et al., 2020). It aims to improve the overall performance of all multiple tasks simultaneously different from the sequential multi-task leaning (Requeima et al., 2019; Garnelo et al., 2018a). This is a challenging scenario due to the domain shift between tasks, which makes it sub-optimal to directly apply neural processes by incorporating the data from other related tasks into the context of each individual task.

### 3.1 HIERARCHICAL CONTEXT MODELING

Multi-task learning considers the estimation of random functions $f_l$, $l = 1, ..., L$ for each of the $L$ related tasks. Each task $l$ has its own training data, which is split into a context set $D_l = (\mathbf{X}_l, \mathbf{Y}_l)$ and a target set $D_l^* = (\mathbf{X}_l^*, \mathbf{Y}_l^*)$. $\mathbf{X}_l^* \in \mathbb{R}^{n_l^* \times d}$ and $\mathbf{X}_l \in \mathbb{R}^{n_l \times d}$ are inputs while $\mathbf{Y}_l^* \in \mathbb{R}^{n_l^* \times C}$ and $\mathbf{Y}_l \in \mathbb{R}^{n_l \times C}$ are outputs. $d$ and $C$ are the sizes of the input space and output space, respectively. $n_l^*$ and $n_l$ are the sizes of respectively the target and context set for the $l$-th task. Thus, we obtain the task-specific latent variables $\mathbf{f}_l = f_l(\mathbf{X}_l) \in \mathbb{R}^{n_l \times C}$. We use $\{D_l\}_{l=1}^L$ to denote all context sets in the dataset, which for brevity we represent as $\{D_l\}$ and likewise will do for other sets.

Formulated this way the goal of multi-task learning is to predict $\{\mathbf{Y}_l^*\}$ for given $\{\mathbf{X}_l^*\}$ simultaneously with the assistance information $\{D_l\}$ from all tasks. To this end, we construct a joint prediction distribution with respect to the latent random functions $\{f_l\}$ as:

$$p(\{\mathbf{Y}_l^*\}|\{\mathbf{X}_l^*\}, \{D_l\}) = \prod_{l=1}^L p(\mathbf{Y}_l^*|\mathbf{X}_l^*, \{D_l\}) = \prod_{l=1}^L \int p(\mathbf{Y}_l^*|\mathbf{X}_l^*, f_l) p(f_l|\mathbf{M}) df_l. \qquad (3)$$

Here, to enable shared knowledge to be transferred among tasks, we introduce a global variable $\mathbf{M}$ which works as a container to collect the useful information from $\{D_l\}$ of all tasks. In contrast to neural processes for single tasks, the global variable $\mathbf{M}$ provides the contextual information from all tasks for each individual task. The concrete formation of $\mathbf{M}$ depends on the learning scenarios. For regression tasks, $\mathbf{M} \in \mathbb{R}^{L*d}$ and each row corresponds to one task, which is the average of all feature vectors from a task. For classification tasks, $\mathbf{M} \in \mathbb{R}^{L*C*d}$ where each vector $\mathbf{M}_{l,c}$ is the average of all features of each category from the corresponding task.

Similar to Gaussian processes, we assume that the function value is $\mathbf{Y}_l^* = f_l(\mathbf{X}_l^*) + \epsilon$, where $\epsilon \sim \mathcal{N}(0, \sigma^2)$ is the observation noise. For regression tasks, we can define the predictive likelihood on the target set as $p(\mathbf{Y}_l^*|\mathbf{X}_l^*, f_l) = \mathcal{N}(\mathbf{Y}_l^*|f_l(\mathbf{X}_l^*), \sigma^2)$. For classification tasks, we use $\log p(\mathbf{Y}_l^*|\mathbf{X}_l^*, f_l) = \sum_{i=1}^{n_l^*} \mathbf{y}_{l,i}^* \log(f_l(\mathbf{x}_{l,i}^*))$ as the log-likelihood function, where $\mathbf{x}_{l,i}^*$ is the $i$-th target sample from the $l$-th task.

In order to combine Gaussian processes and neural networks in the context of multi-task learning, we define $p(f_l|\mathbf{M})$ in (3) as a deep neural network in place of a Gaussian distribution parameterized by a kernel function. To be more specific, we assume that $f_l$ is parameterized by a random variable $\psi_l$ by defining $f_l(\mathbf{X}) = \mathbf{X}\psi_l^\top$. We specify a data dependent prior by conditioning $\psi_l$ on the global variable $\mathbf{M}$. In this way, we incorporate the transferable knowledge into the learning of the prediction function of the current tasks. Thus, the predictive distribution for the $l$-th task over its target set can be formulated as follows:

$$p(\mathbf{Y}_l^*|\mathbf{X}_l^*, \{D_l\}) = \int p(\mathbf{Y}_l^*|\mathbf{X}_l^*, \boldsymbol{\psi}_l)p_\theta(\boldsymbol{\psi}_l|\mathbf{M})d\boldsymbol{\psi}_l. \tag{4}$$

In effect, $\boldsymbol{\psi}_l$ denotes the parameters of classifier for classification tasks or regressors for regression tasks. Particularly, the introduced latent variable $\boldsymbol{\psi}_l \in \mathbb{R}^{C \times d}$ denotes the task specific classifier, where $d$ is the dimension of the input feature and $C$ is the number of classes in the dataset. As done in (Requeima et al., 2019), we generate each column of $\boldsymbol{\psi}_l$ independently from the context samples of the corresponding class. In our case, each column of $\boldsymbol{\psi}_l$ encodes the context information of its class from all tasks $p_\theta(\boldsymbol{\psi}_l|\mathbf{M}) = \prod_{c=1}^C p_\theta(\boldsymbol{\psi}_{l,c}|\mathbf{M}_c)$.

Directly aggregating $\mathbf{M}$ as done in neural processes for single tasks is not applicable for multi-task learning due to the distribution shift between tasks. The data from related tasks should be processed and adapted to the current task as the contextual information. To this end, we introduce a higher-level latent variable $\boldsymbol{\alpha}_l$ to extract the shared knowledge from $\mathbf{M}$, which is conditioned on the data $D_l$ of each task. This results in a hierarchical Bayesian modeling of functions in the neural process:

$$p_\theta(\boldsymbol{\psi}_l|\mathbf{M}) = \int p_{\theta_1}(\boldsymbol{\psi}_l|\boldsymbol{\alpha}_l, \mathbf{M})p_{\theta_2}(\boldsymbol{\alpha}_l|D_l)d\boldsymbol{\alpha}_l, \tag{5}$$

where $\boldsymbol{\alpha}_l$ is the latent variable to control the access to shared knowledge for each task, which is used to explore the relevant knowledge to the task $l$. $p_{\theta_1}(\boldsymbol{\psi}_l|\boldsymbol{\alpha}_l, \mathbf{M})$ and $p_{\theta_2}(\boldsymbol{\alpha}_l|D_l)$ are prior distributions of the latent variable $\boldsymbol{\psi}_l$ and $\boldsymbol{\alpha}_l$, respectively, which are parameterized with neural networks. To be specific, we define $p_{\theta_1}(\boldsymbol{\psi}_l|\boldsymbol{\alpha}_l, \mathbf{M}) = \mathcal{N}(\boldsymbol{\psi}_l|\mu(\mathbf{m}_l), \Sigma(\mathbf{m}_l))$. Here $\mathbf{m}_l$ contains the relevant knowledge to the task $l$, which is adapted from the global variable $\mathbf{M}$ by a deterministic function $\mathbf{m}_l = h(\boldsymbol{\alpha}_l, \mathbf{M})$, where $h(\cdot)$ is a learnable function implemented with a neural network.

By substituting (5) into (4), we obtain the model of multi-task neural processes as follows:

$$p(\{\mathbf{Y}_l^*\}|\{\mathbf{X}_l^*\}, \{D_l\}) = \prod_{l=1}^L \int \int p(\mathbf{Y}_l^*|\mathbf{X}_l^*, \boldsymbol{\psi}_l)p_{\theta_1}(\boldsymbol{\psi}_l|\boldsymbol{\alpha}_l, \mathbf{M})p_{\theta_2}(\boldsymbol{\alpha}_l|D_l)d\boldsymbol{\psi}_l d\boldsymbol{\alpha}_l. \tag{6}$$

The designed hierarchical context modeling provides a principled way to explore task relatedness in the function space, which allows task-specific function variables to leverage the shared knowledge from related tasks. We provide theoretical proof in Appendix to show that the proposed multi-task neural processes are a valid stochastic processes, which completes the theory of multi-task neural processes. The graphical model of the multi-task neural processes is shown in Figure 1 (b).

## 3.2 Variational Hierarchical Inference

The previous section developed the model of multi-tasks neural processes with a hierarchical context model. We now describe how to optimize the model and obtain the predictions by leveraging a variational Bayesian inference framework. To that end, based on a conditional independence assumption, we introduce the variational joint posterior distribution factorized as follows:

$$q_\varphi(\{\boldsymbol{\psi}_l\}, \{\boldsymbol{\alpha}_l\}|\{D_l^*\}) = \prod_{l=1}^L q_\varphi(\boldsymbol{\psi}_l, \boldsymbol{\alpha}_l|D_l^*) = \prod_{l=1}^L q_{\varphi_1}(\boldsymbol{\psi}_l|D_l^*)q_{\varphi_2}(\boldsymbol{\alpha}_l|D_l^*), \tag{7}$$

where $q_{\varphi_1}(\boldsymbol{\psi}_l|D_l^*)$ and $q_{\varphi_2}(\boldsymbol{\alpha}_l|D_l^*)$ are variational posteriors of the latent variables $\boldsymbol{\psi}_l$ and $\boldsymbol{\alpha}_l$ for the task $l$, respectively. Both variational posteriors are parameterized as Gaussian distributions. $\varphi_1$ and $\varphi_2$ are amortized inference networks to generate variational posteriors and shared by all tasks. We make use of the amortized variational inference technique (Kingma & Welling, 2013) to learn such distributions over latent variables.

**Learning** By incorporating the variational posteriors into (7), we derive the ELBO for the multi-task neural processes as follows:

$$\log p(\{\mathbf{Y}_l^*\}|\{\mathbf{X}_l^*\}, \{D_l\}) \geq \sum_{l=1}^L \Big\{ \mathbb{E}_{q_{\varphi_2}(\boldsymbol{\alpha}_l|D_l^*)}\big\{\mathbb{E}_{q_{\varphi_1}(\boldsymbol{\psi}_l|D_l^*)}[\log p(\mathbf{Y}_l^*|\mathbf{X}_l^*, \boldsymbol{\psi}_l)]$$
$$- \mathbb{D}_{\mathrm{KL}}[q_{\varphi_1}(\boldsymbol{\psi}_l|D_l^*)||p_{\theta_1}(\boldsymbol{\psi}_l|\boldsymbol{\alpha}_l, \mathbf{M})]\big\} - \mathbb{D}_{\mathrm{KL}}[q_{\varphi_2}(\boldsymbol{\alpha}_l|D_l^*)||p_{\theta_2}(\boldsymbol{\alpha}_l|D_l)] \Big\}. \tag{8}$$

The detailed derivation is provided in Appendix A. By adopting the Monte Carlo sampling, we obtain the empirical objective for the proposed multi-task neural processes:

$$
\hat{L}_{\mathrm{MTNPs}}(\theta, \varphi) = \sum_{l=1}^{L} \Big\{ \frac{1}{N_a} \sum_{i=1}^{N_a} \Big\{ \frac{1}{N_f} \sum_{j=1}^{N_f} [-\log p(\mathbf{Y}_l^* | \mathbf{X}_l^*, \boldsymbol{\psi}_l^{(j)})]
$$
$$
+ \lambda_f \mathbb{D}_{\mathrm{KL}}[q_{\varphi_1}(\boldsymbol{\psi}_l | D_l^*) || p_{\theta_1}(\boldsymbol{\psi}_l | \boldsymbol{\alpha}_l^{(i)}, \mathbf{M})] \Big\} + \lambda_a \mathbb{D}_{\mathrm{KL}}[q_{\varphi_2}(\boldsymbol{\alpha}_l | D_l^*) || p_{\theta_2}(\boldsymbol{\alpha}_l | D_l)] \Big\},
$$
$$(9)$$

where $\boldsymbol{\psi}_l^{(j)} \sim q_{\varphi_1}(\boldsymbol{\psi}_l | D_l^*)$ and $\boldsymbol{\alpha}_l^{(i)} \sim q_{\varphi_2}(\boldsymbol{\alpha}_l | D_l^*)$. $N_f$ and $N_a$ are the number of Monte Carlo samples for the variational posteriors of $\boldsymbol{\psi}$ and $\boldsymbol{\alpha}$, respectively. $\lambda_f$ and $\lambda_a$ are the hyperparameters to help stably train the KL-divergence terms, which are set following the annealing scheme of (Bowman et al., 2015). In practice, we apply the local reparameterization trick (Kingma et al., 2015) to reduce the variance of stochastic gradients.

**Prediction** Having the learned model, we can make prediction on the target set. Given a test sample $\mathbf{x}_l$ from the $l$-th task, we can produce the predictive distribution which involves the prior distributions $p_{\theta_1}(\boldsymbol{\psi}_l | \boldsymbol{\alpha}_l, \mathbf{M})$ and $p_{\theta_2}(\boldsymbol{\alpha}_l | D_l)$. The predictive distribution with the integration for the introduced latent variables is formulated as:

$$
p(\mathbf{y}_l | \mathbf{x}_l, \{D_l\}) = \int \int p(\mathbf{y}_l | \mathbf{x}_l, \boldsymbol{\psi}_l) p_{\theta_1}(\boldsymbol{\psi}_l | \boldsymbol{\alpha}_l, \mathbf{M}) p_{\theta_2}(\boldsymbol{\alpha}_l | D_l) d\boldsymbol{\alpha}_l d\boldsymbol{\psi}_l \qquad (10)
$$

Here we also need to apply the Monte Carlo estimation over (10) and obtain the predictions as follows:

$$
p(\mathbf{y}_l | \mathbf{x}_l) \approx \frac{1}{N_a} \sum_{i=1}^{N_a} \frac{1}{N_f} \sum_{j=1}^{N_f} p(\mathbf{y}_l | \mathbf{x}_l, \boldsymbol{\psi}_l^{(j)})], \qquad (11)
$$

where $\boldsymbol{\psi}_l^{(j)} \sim p_{\theta_1}(\boldsymbol{\psi}_l | \boldsymbol{\alpha}_l^{(i)}, \mathbf{M})$ and $\boldsymbol{\alpha}_l^{(i)} \sim p_{\theta_2}(\boldsymbol{\alpha}_l | D_l)$. In particularly, the prior distribution of the latent variable $\boldsymbol{\psi}_l$ is generated from the output of the learned function $h(\boldsymbol{\alpha}_l^{(i)}, \mathbf{M})$.

## 4 RELATED WORK

Neural network based models have achieved impressive results on various applications (LeCun et al., 2015). However, due to the large number of parameters these neural models need large-scale data with annotation. It is challenging to train a deep neural model that generalizes well with number-limited labeled data. To reduce such labeling consumption, recent works (Long et al., 2017; Liu et al., 2016) follow a multi-task learning strategy to fully leverage information from relevant tasks as inductive bias (Caruana, 1997) to improve each task's performance. The main challenge of multi-task learning is that each task is provided with a limited amount of labeled data, which is insufficient to build reliable classifiers without overfitting (Long et al., 2017).

Multi-task learning (MTL) aims to learn several tasks simultaneously and improve their overall performance. The crux of MTL is how to explore task relatedness from different tasks, which could be particularly significant when limited data for each task is available (Long et al., 2017; Zhang et al., 2020). Recently, the task relatedness is learned in many different aspects of the model, e.g., loss functions (Liu et al., 2020; Qian et al., 2020; Kendall et al., 2018), parameter space (Long et al., 2017; Bakker & Heskes, 2003), or representation space (Misra et al., 2016). In this paper, we focus on the data-insufficient problem for multi-task learning. Different from other branches of MTL models (Huang et al., 2021; Fu et al., 2021; Phillips et al., 2021) which leverage several channels of supervision information simultaneously included by the same input, our data setting allows each task to have its own individual data. This is much more challenging due to the distribution shift between inputs of different tasks.

In previous works (Lawrence & Platt, 2004; Yu et al., 2005; Yousefi et al., 2019), multi-task learning benefits from Gaussian Processes by generally incorporating the shared information in the Gaussian Processes prior to synergize several random functions of different tasks simultaneously. The multi-task Gaussian processes (Yu et al., 2005) proposes a hierarchical prior which enables task specific random functions to share some common structure in the hyper prior. As an alternative to using

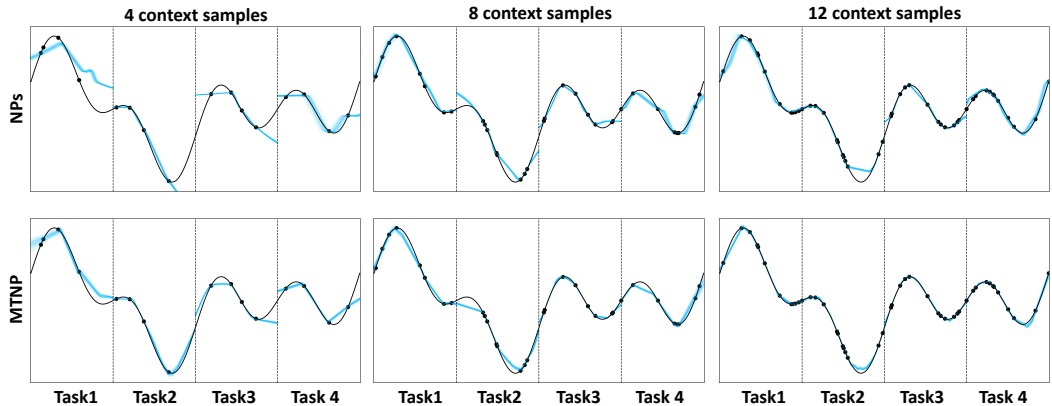

Figure 2: 1-D multi-task function regression. The plots show sample of curves conditioned on an increasing number of context samples for each task (4, 8 and 12 in each column respectively). The ground truth is shown in black and the context samples as black dots. The predictions of our MTNPs more resemble the ground truth than that of NPs, especially in the boundary of different tasks.

Gaussian processes for multi-task learning, nowadays deep neural networks have become popular as parameterized functions, which construct an information sharing architecture between tasks. Some methods (Misra et al., 2016; Liu et al., 2019) learn expressive combinations of features from different tasks by deep neural networks. Some approaches (Gao et al., 2020; Sun et al., 2019) flexibly adjust the deep architecture for each individual tasks. Yu et al. (2020) mitigates gradient interference by performing the proposed gradient surgery. However, these deep multi-task learning methods relies on large amounts of training data and therefore tends to overfit with limited data.

Recently, neural processes (Garnelo et al., 2018b) and the related works (Garnelo et al., 2018a; Kim et al., 2019; Wang & Van Hoof, 2020; Requeima et al., 2019; Gordon et al., 2019) combine Gaussian Processes and neural networks, which are not only computationally efficient but also retain a probabilistic interpretation of the model. (Garnelo et al., 2018b; Kim et al., 2019; Wang & Van Hoof, 2020) introduce the latent representation variables to model the function randomness. (Requeima et al., 2019; Gordon et al., 2019) introduce a latent parameter variable to directly model the predictive distribution, which acts as the parameters of the neural network.

Hierarchical modeling in the Bayesian framework has been successful to design the form of the prior (Daumé III, 2009; Zhao et al., 2017; Klushyn et al., 2019; Wang & Van Hoof, 2020) and posterior distributions (Ranganath et al., 2016; Krueger et al., 2017; Zhen et al., 2020) based on many observations. It allows the latent variable to follow a complicated distribution and forms a highly flexible approximation (Krueger et al., 2017).

## 5  EXPERIMENTS AND RESULTS

### 5.1  1-D MULTI-TASK FUNCTION REGRESSION

Our method fully utilizes the transferable knowledge provided by related tasks to improve each task's performance. To show this, we test the proposed multi-task neural processes on the 1-D multi-task function regression.

**Setup.**  We define several tasks with different data distributions: inputs of each task are sampled from the separated intervals without overlapping, such as $[-2\pi, \pi)$, $[-\pi, 0)$, $[0, \pi)$, and $[\pi, 2\pi)$. Each $x$-value is drawn uniformly at random in its belonging intervals. We assume that all tasks share the same ground truth function to ensure that there is transferable knowledge between them. The ground truth function is characterized as a sum of sine or cosine functions (Wang & Van Hoof, 2020), such as $y = \sin(x) + \sin(2x) - \cos(0.5x) + \epsilon$, where $\epsilon$ is the noise drawn from $\mathcal{N}(0, 0.0003^2)$. At each training step, the hyperparameters of the ground truth function are fixed as set in (Kim et al., 2019; Wang & Van Hoof, 2020).

**Results.**  As shown in Figure 2, we visualize prediction results of different tasks for the comparison of neural processes and multi-task neural processes. The predictions of the proposed multi-task neural processes (the second row) more resemble the ground truth functions than that of neural processes (the first row), especially in the boundary of different tasks. Moreover, we find that when training with fewer context samples, the improvement is more significant. This demonstrates that our method enhances the generalization of each task by fully utilizing the transferable knowledge from related tasks.

## 5.2 MULTI-TASK CLASSIFICATION

**Datasets.**  We evaluate the performance of our multi-task neural processes on real-world multi-task classification, where different tasks are defined as image classification problems in different domains. The tasks are related to each other since they share the same label space. `Office-Home` (Venkateswara et al., 2017) contains images from four domains/tasks: Artistic (A), Clipart (C), Product (P) and Real-world (R). Each task contains images from 65 categories collected under office and home settings. There are about $15,500$ images in total. `Office-Caltech` (Gong et al., 2012) contains the ten categories shared between Office-31 (Saenko et al., 2010) and Caltech-256 (Griffin et al., 2007). One task uses data from Caltech-256 (C), and the other three tasks use data from Office-31, whose images were collected from three distinct domains/tasks, namely Amazon (A), Webcam (W) and DSLR (D). There are $8 \sim 151$ samples per category per task, and $2,533$ images in total. `ImageCLEF` (Long et al., 2017), the benchmark for the ImageCLEF domain adaptation challenge, contains 12 common categories shared by four public datasets/tasks: Caltech-256 (C), ImageNet ILSVRC 2012 (I), Pascal VOC 2012 (P), and Bing (B). There are $2,400$ images in total.

**Setup.**  We adopt the standard evaluation protocols (Zhang & Yang, 2021) for multi-task classification datasets. We randomly select $5\%$, $10\%$ and $20\%$ labeled data for training, which correspond to about 3, 6 and 12 samples per category per task, respectively. In this case, each task has a limited amount of training data, which is insufficient for building the reliable classifier without overfitting. For all three benchmarks, we extract the input features by the pretrained VGGnet as (Long et al., 2017). The architectures of inference networks used in our model are provided in Appendix C. All the results with error bars are obtained based on a 95% confidence interval from five runs.

**Compared methods.**  To show the effectiveness of the proposed multi-tasks neural processes, we conduct a thorough comparison implementing multiple different baseline models. Single task learning (STL) is implemented by task-specific feature extractors and classifiers without knowledge sharing among tasks. Basic multi-task learning (BMTL) shares feature extractors and adds task specific classifiers. We also define variational extensions of the single task learning (VSTL) and basic multi-task learning (VBMTL), which cast models as variational Bayesian problems and treat classifiers as latent variables (Shen et al., 2021). We implement neural processes (NPs) and its variant, neural processes with all task context for comparison. NPs with all task context is a straightforward extension of NPs for MTL, which treats context sets from all tasks equally without hierarchical context modeling. For a fair comparison, all the above-mentioned methods share the same architecture of the feature extractor.

**Results.**  We provide comprehensive comparisons on `Office-Home` in Table 1, which is a more challenging multi-task classification dataset with 65 categories. The results show that our MTNPs outperform other counterpart methods. NPs with all task context performs better than NPs by a large margin when $5\%$ labeled data is available, showing the benefit of exploring shared knowledge from related tasks with limited data. Noticeably compared to NPs with all task context, the proposed multi-task neural processes benefit from hierarchical context modeling and show even better performance.

Table 1: Performance comparison (average accuracy) on `Office-Home` training with $5\%$, $10\%$, $20\%$ labeled data. The higher the better.

| Methods | 5% | 10% | 20% |
|---|---|---|---|
| STL | $49.2_{\pm 0.2}$ | $58.3_{\pm 0.1}$ | $64.9_{\pm 0.1}$ |
| VSTL | $51.1_{\pm 0.1}$ | $60.2_{\pm 0.2}$ | $65.8_{\pm 0.2}$ |
| BMTL | $50.4_{\pm 0.1}$ | $59.5_{\pm 0.1}$ | $65.6_{\pm 0.1}$ |
| VBMTL | $51.3_{\pm 0.1}$ | $60.9_{\pm 0.1}$ | $67.0_{\pm 0.2}$ |
| NPs | $54.7_{\pm 0.1}$ | $59.4_{\pm 0.2}$ | $69.3_{\pm 0.2}$ |
| NPs with all task context | $59.1_{\pm 0.2}$ | $62.4_{\pm 0.1}$ | $69.1_{\pm 0.1}$ |
| MTNPs | $\mathbf{60.0_{\pm 0.1}}$ | $\mathbf{63.3_{\pm 0.1}}$ | $\mathbf{69.9_{\pm 0.3}}$ |

Table 2: Performance comparison of different methods on the `Office-Home` dataset.

| Methods | 5% | | | | | 10% | | | | |
|---|---|---|---|---|---|---|---|---|---|---|
| | A | C | P | R | Avg. | A | C | P | R | Avg. |
| STL | $36.7_{\pm0.4}$ | $30.8_{\pm0.5}$ | $67.5_{\pm0.3}$ | $61.7_{\pm0.3}$ | $49.2_{\pm0.2}$ | $50.4_{\pm0.3}$ | $40.8_{\pm0.3}$ | $74.4_{\pm0.4}$ | $67.5_{\pm0.4}$ | $58.3_{\pm0.1}$ |
| Bakker & Heskes (2003) | $40.0_{\pm0.1}$ | $33.6_{\pm0.3}$ | $69.8_{\pm0.4}$ | $63.6_{\pm0.3}$ | $52.8_{\pm0.1}$ | $52.5_{\pm0.3}$ | $42.3_{\pm0.4}$ | $75.7_{\pm0.5}$ | $69.5_{\pm0.5}$ | $60.0_{\pm0.2}$ |
| Long et al. (2017) | $47.8_{\pm0.4}$ | $37.9_{\pm0.2}$ | $73.6_{\pm0.3}$ | $70.4_{\pm0.2}$ | $57.4_{\pm0.1}$ | $57.2_{\pm0.3}$ | $43.3_{\pm0.1}$ | $78.7_{\pm0.3}$ | $74.4_{\pm0.1}$ | $\mathbf{63.4_{\pm0.2}}$ |
| Kendall et al. (2018) | $40.2_{\pm0.2}$ | $33.6_{\pm0.4}$ | $69.5_{\pm0.2}$ | $63.7_{\pm0.1}$ | $51.8_{\pm0.1}$ | $49.1_{\pm0.1}$ | $38.7_{\pm0.3}$ | $73.4_{\pm0.2}$ | $67.4_{\pm0.3}$ | $57.2_{\pm0.2}$ |
| Guo et al. (2020) | $25.8_{\pm1.4}$ | $26.7_{\pm0.8}$ | $55.8_{\pm0.7}$ | $46.0_{\pm0.6}$ | $38.3_{\pm0.5}$ | $38.3_{\pm0.9}$ | $41.5_{\pm0.8}$ | $67.6_{\pm0.4}$ | $58.8_{\pm0.1}$ | $51.5_{\pm0.3}$ |
| Qian et al. (2020) | $37.9_{\pm0.3}$ | $31.4_{\pm0.2}$ | $67.7_{\pm0.3}$ | $62.4_{\pm0.2}$ | $49.9_{\pm0.2}$ | $47.1_{\pm0.2}$ | $37.2_{\pm0.1}$ | $70.5_{\pm0.2}$ | $66.3_{\pm0.3}$ | $55.3_{\pm0.1}$ |
| MTNPs | $55.0_{\pm0.2}$ | $40.8_{\pm0.3}$ | $74.2_{\pm0.2}$ | $69.9_{\pm0.2}$ | $\mathbf{60.0_{\pm0.1}}$ | $59.5_{\pm0.3}$ | $44.4_{\pm0.5}$ | $77.3_{\pm0.3}$ | $72.0_{\pm0.3}$ | $63.3_{\pm0.1}$ |

Table 3: Performance comparison of different methods on the `Office-Caltech` dataset.

| Methods | 5% | | | | | 10% | | | | |
|---|---|---|---|---|---|---|---|---|---|---|
| | A | W | D | C | Avg. | A | W | D | C | Avg. |
| STL | $87.4_{\pm0.4}$ | $87.9_{\pm0.3}$ | $96.4_{\pm0.5}$ | $82.8_{\pm0.2}$ | $88.6_{\pm0.3}$ | $92.8_{\pm0.5}$ | $97.7_{\pm0.3}$ | $87.8_{\pm0.2}$ | $84.3_{\pm0.4}$ | $90.7_{\pm0.2}$ |
| Bakker & Heskes (2003) | $93.2_{\pm0.2}$ | $94.0_{\pm0.3}$ | $94.7_{\pm0.3}$ | $85.4_{\pm0.4}$ | $91.8_{\pm0.1}$ | $94.9_{\pm0.4}$ | $97.6_{\pm0.5}$ | $96.6_{\pm0.5}$ | $90.9_{\pm0.4}$ | $95.0_{\pm0.2}$ |
| Long et al. (2017) | $92.7_{\pm0.2}$ | $94.3_{\pm0.2}$ | $97.1_{\pm0.2}$ | $89.2_{\pm0.6}$ | $93.4_{\pm0.2}$ | $95.0_{\pm0.3}$ | $98.1_{\pm0.4}$ | $95.0_{\pm0.5}$ | $91.3_{\pm0.2}$ | $94.8_{\pm0.3}$ |
| Kendall et al. (2018) | $93.6_{\pm0.4}$ | $92.5_{\pm0.2}$ | $95.0_{\pm0.5}$ | $83.9_{\pm0.5}$ | $91.2_{\pm0.3}$ | $94.9_{\pm0.5}$ | $96.2_{\pm0.4}$ | $93.6_{\pm0.3}$ | $90.4_{\pm0.2}$ | $93.8_{\pm0.2}$ |
| Guo et al. (2020) | $73.9_{\pm2.5}$ | $76.0_{\pm3.6}$ | $78.3_{\pm1.2}$ | $70.3_{\pm0.9}$ | $74.6_{\pm0.9}$ | $80.4_{\pm1.8}$ | $89.4_{\pm2.9}$ | $73.4_{\pm4.4}$ | $78.5_{\pm1.9}$ | $80.4_{\pm1.2}$ |
| Qian et al. (2020) | $92.6_{\pm0.3}$ | $90.9_{\pm0.2}$ | $95.7_{\pm0.4}$ | $85.2_{\pm0.6}$ | $91.1_{\pm0.3}$ | $94.2_{\pm0.4}$ | $97.0_{\pm0.4}$ | $95.0_{\pm0.4}$ | $90.2_{\pm0.3}$ | $94.1_{\pm0.3}$ |
| MTNPs | $94.6_{\pm0.1}$ | $95.8_{\pm0.2}$ | $97.9_{\pm0.0}$ | $90.2_{\pm0.1}$ | $\mathbf{94.6_{\pm0.1}}$ | $95.1_{\pm0.1}$ | $97.7_{\pm0.1}$ | $97.1_{\pm0.3}$ | $91.6_{\pm0.3}$ | $\mathbf{95.4_{\pm0.1}}$ |

Table 4: Performance comparison of different methods on the `ImageCLEF` dataset.

| Methods | 5% | | | | | 10% | | | | |
|---|---|---|---|---|---|---|---|---|---|---|
| | C | I | P | B | Avg. | C | I | P | B | Avg. |
| STL | $85.4_{\pm0.6}$ | $71.4_{\pm0.4}$ | $57.7_{\pm0.2}$ | $36.0_{\pm0.2}$ | $62.6_{\pm0.2}$ | $88.9_{\pm0.5}$ | $77.8_{\pm0.3}$ | $64.3_{\pm0.2}$ | $47.6_{\pm0.5}$ | $69.7_{\pm0.3}$ |
| Bakker & Heskes (2003) | $90.9_{\pm0.4}$ | $85.4_{\pm0.6}$ | $68.1_{\pm0.3}$ | $51.4_{\pm0.5}$ | $73.9_{\pm0.3}$ | $91.0_{\pm0.5}$ | $87.1_{\pm0.3}$ | $73.4_{\pm0.4}$ | $54.5_{\pm0.2}$ | $76.5_{\pm0.4}$ |
| Long et al. (2017) | $90.1_{\pm0.5}$ | $76.5_{\pm0.5}$ | $72.8_{\pm0.3}$ | $54.9_{\pm0.4}$ | $73.7_{\pm0.4}$ | $93.3_{\pm0.4}$ | $83.2_{\pm0.6}$ | $70.4_{\pm0.4}$ | $56.3_{\pm0.4}$ | $75.8_{\pm0.2}$ |
| Kendall et al. (2018) | $93.2_{\pm0.6}$ | $86.1_{\pm0.4}$ | $68.6_{\pm0.3}$ | $50.4_{\pm0.4}$ | $74.6_{\pm0.2}$ | $91.9_{\pm0.3}$ | $88.9_{\pm0.5}$ | $74.3_{\pm0.3}$ | $52.4_{\pm0.2}$ | $76.9_{\pm0.3}$ |
| Guo et al. (2020) | $80.1_{\pm2.9}$ | $55.5_{\pm1.2}$ | $46.7_{\pm1.1}$ | $24.4_{\pm1.3}$ | $51.7_{\pm0.9}$ | $86.1_{\pm1.6}$ | $68.9_{\pm2.3}$ | $56.0_{\pm1.5}$ | $39.3_{\pm2.7}$ | $62.6_{\pm0.8}$ |
| Qian et al. (2020) | $91.6_{\pm0.3}$ | $85.8_{\pm0.4}$ | $68.4_{\pm0.3}$ | $50.2_{\pm0.4}$ | $74.0_{\pm0.4}$ | $90.7_{\pm0.4}$ | $88.1_{\pm0.6}$ | $75.6_{\pm0.4}$ | $54.6_{\pm0.3}$ | $77.3_{\pm0.3}$ |
| MTNPs | $90.5_{\pm0.3}$ | $84.9_{\pm0.2}$ | $70.2_{\pm0.2}$ | $58.9_{\pm0.4}$ | $\mathbf{76.1_{\pm0.1}}$ | $93.5_{\pm0.4}$ | $88.5_{\pm0.3}$ | $74.6_{\pm0.4}$ | $61.7_{\pm0.3}$ | $\mathbf{79.6_{\pm0.1}}$ |

Table 5: Performance of multi-task regression (normalized mean squared errors) for rotation angle estimation. The lower the better.

| Methods | 0 | 1 | 2 | 3 | 4 | 5 | 6 | 7 | 8 | 9 | Avg. |
|---|---|---|---|---|---|---|---|---|---|---|---|
| STL | $.138_{\pm.003}$ | $.158_{\pm.003}$ | $.245_{\pm.015}$ | $.216_{\pm.023}$ | $.327_{\pm.019}$ | $.150_{\pm.002}$ | $.229_{\pm.010}$ | $.286_{\pm.032}$ | $.209_{\pm.018}$ | $.191_{\pm.006}$ | $.215_{\pm.001}$ |
| VSTL | $.144_{\pm.002}$ | $.161_{\pm.012}$ | $.330_{\pm.018}$ | $.197_{\pm.006}$ | $.382_{\pm.028}$ | $.208_{\pm.016}$ | $.189_{\pm.002}$ | $.296_{\pm.024}$ | $.135_{\pm.004}$ | $.194_{\pm.003}$ | $.224_{\pm.004}$ |
| BMTL | $.114_{\pm.004}$ | $.124_{\pm.005}$ | $.115_{\pm.004}$ | $.124_{\pm.006}$ | $.115_{\pm.004}$ | $.114_{\pm.004}$ | $.114_{\pm.003}$ | $.125_{\pm.005}$ | $.115_{\pm.004}$ | $.115_{\pm.004}$ | $.118_{\pm.003}$ |
| VBMTL | $.121_{\pm.003}$ | $.124_{\pm.005}$ | $.121_{\pm.003}$ | $.123_{\pm.005}$ | $.121_{\pm.003}$ | $.121_{\pm.003}$ | $.121_{\pm.003}$ | $.124_{\pm.005}$ | $.121_{\pm.003}$ | $.121_{\pm.003}$ | $.121_{\pm.003}$ |
| Yu et al. (2005) | $.171_{\pm.004}$ | $.154_{\pm.002}$ | $.145_{\pm.001}$ | $.126_{\pm.002}$ | $.168_{\pm.002}$ | $.163_{\pm.001}$ | $.224_{\pm.003}$ | $.145_{\pm.002}$ | $.113_{\pm.002}$ | $.122_{\pm.002}$ | $.153_{\pm.000}$ |
| Liu et al. (2019) | $.196_{\pm.020}$ | $.096_{\pm.110}$ | $.162_{\pm.032}$ | $.124_{\pm.015}$ | $.152_{\pm.049}$ | $.140_{\pm.025}$ | $.249_{\pm.045}$ | $.195_{\pm.016}$ | $.081_{\pm.018}$ | $.119_{\pm.021}$ | $.152_{\pm.018}$ |
| Guo et al. (2020) | $.158_{\pm.002}$ | $.078_{\pm.004}$ | $.103_{\pm.004}$ | $.063_{\pm.003}$ | $.118_{\pm.008}$ | $.099_{\pm.004}$ | $.156_{\pm.004}$ | $.090_{\pm.006}$ | $.082_{\pm.004}$ | $.138_{\pm.009}$ | $.109_{\pm.002}$ |
| NPs | $.193_{\pm.001}$ | $.058_{\pm.003}$ | $.105_{\pm.003}$ | $.067_{\pm.004}$ | $.101_{\pm.002}$ | $.120_{\pm.003}$ | $.158_{\pm.004}$ | $.107_{\pm.003}$ | $.083_{\pm.005}$ | $.126_{\pm.003}$ | $.112_{\pm.003}$ |
| NPs with all task context | $.188_{\pm.002}$ | $.064_{\pm.003}$ | $.114_{\pm.003}$ | $.063_{\pm.005}$ | $.103_{\pm.001}$ | $.116_{\pm.003}$ | $.167_{\pm.003}$ | $.095_{\pm.004}$ | $.067_{\pm.001}$ | $.111_{\pm.002}$ | $.109_{\pm.002}$ |
| MTNPs | $.183_{\pm.002}$ | $.060_{\pm.003}$ | $.098_{\pm.002}$ | $.067_{\pm.001}$ | $.109_{\pm.003}$ | $.109_{\pm.002}$ | $.160_{\pm.004}$ | $.092_{\pm.002}$ | $.077_{\pm.003}$ | $.113_{\pm.004}$ | $\mathbf{.106_{\pm.001}}$ |

More comparison results on the `Office-Home`, `Office-Caltech`, `ImageCLEF` datasets are shown in Tables 2, 3 and 4, respectively. The average accuracy of all tasks is used for overall performance measurement. The best results are marked in bold. Our MTNPs achieve competitive and even better performance on such multi-task classification datasets with different train-test splits. Compared with Bayesian baselines, including VSTL, VBMTL, NPs and (Bakker & Heskes, 2003), our MTNPs directly infer the parameters of prediction functions rather than the input representation, which is able to model a broader range of functional distribution. Moreover, in function space the hierarchical context modeling can better explore the task relatedness, which enable the models to capture the relevant knowledge even in presence of distribution shift among tasks. Experimental results on all three benchmarks with 20% labeled data are provided in Appendix D.

## 5.3 MULTI-TASK REGRESSION

**Setup.** In order to show the effectiveness of MTNPs for multi-task regression, we conduct experiments on the `Rotated MNIST` dataset (LeCun et al., 1998). We adopt this dataset to study multi-task regression, where each task is an angle estimation problem for each digit and different tasks corresponding to different digits are related because they share the same rotation angle space. Each image is rotated by $0°$ through $90°$ in intervals of $10°$, where the rotation angle is the regression target. We randomly choose $0.1\%$ samples per task per angle as the training set.

**Results.** Since we would like to improve the overall performance of all regression tasks, we use the average of normalized mean squared errors of all tasks as the measurement. As shown in Table 5, our MTNPs outperform other counterpart methods by yielding an overall lower mean error.

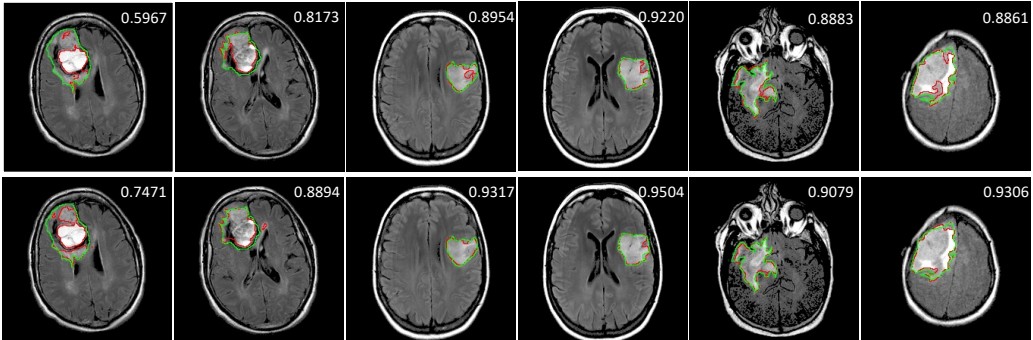

Figure 3: The segmentation results by the proposed multi-task neural processes (bottom row) and the U-Net (upper row). Green outline corresponds to the ground truth and red to the segmentation output. The numbers are the DSC scores compared against ground truth. Our multi-task neural processes can predict contours closer to the ground truth ones with higher DSC scores than U-Net, indicating the advantages of exploring spatial context information for brain image segmentation.

## 5.4 BRAIN IMAGE SEGMENTATION

In this section, we demonstrate that multi-task neural processes are also able to explore spatial context information to improve image segmentation. To this end, we adopt a brain image dataset (Buda et al., 2019) with lower-grade gliomas collected from 110 patients. The number of images varies among patients from 20 to 88. The goal is to segment the tumor in each brain image.

**Setup.**   To apply our multi-task neural processes, we reformulate the segmentation task as a pixel-wise regression problem, where each pixel corresponds to a regression task to predict the probability of this pixel belonging to the tumor. In doing so, the spatial correlation and dependency among pixels are effectively modeled by capturing the task relatedness. To be specific, we consider the prediction of each pixel to be a regression task. For the task $l$, we define $\Omega_l$ as a local region centered at the spatial position $l$, which provides the local context information. In this case, the region centered at the pixel provides the local context information. Each task incorporates the shared knowledge provided by related tasks into its context of the prediction function. This offers an effective way to model the long-range interdependence of pixels in one image. For implementation, we use the U-Net architecture (Ronneberger et al., 2015) as the backbone and add our model as the final layer.

**Results.**   We compare our method and U-Net on the brain segmentation dataset. The results show that the proposed multi-task neural processes surpass the baseline U-Net by 0.5% in terms of dice similarity coefficients (DSC) for the overall validation set. Figure 3 shows segmentation results of the proposed multi-task neural processes (bottom row) and the U-Net (upper row), where the green outline corresponds to the ground truth and the red to the segmentation output. Our multi-task neural processes predict contours closer to the ground truth. This demonstrates the advantages of exploring context information by multi-task neural processes for segmentation.

## 6 CONCLUSION

In this paper, we develop multi-task neural processes, a new variant of neural processes for multi-task learning. We propose to explore the task relatedness in the function space by specifying the function priors in a hierarchical Bayesian inference framework. The shared knowledge from related tasks is incorporated into the context of each individual task, which serves as the inductive bias for making predictions of this task. The hierarchical architecture allows us to design expressive data-dependent prior, enabling the model to explore the complex task relationships in multi-task learning. By leveraging the hierarchical modeling, multi-tasks neural processes are capable of capturing the shared knowledge from other tasks in a principled way by specifying the function prior. We evaluate multi-task neural processes on multi-task regression and classification datasets. Results demonstrate the effectiveness of multi-task neural processes in transferring useful knowledge among tasks for multi-task learning.

## ETHICS STATEMENT

Since our method is a general contribution to the multi-task learning field, it has the potential to benefit any societal problem where multiple tasks are performed simultaneously, e.g., medical diagnosis. Accordingly, the method would also potentially face some negative social impacts accompanying with these applications, e.g., lack of fairness with the model trained by incomplete data due to the privacy of patients in medical imaging tasks.

## REPRODUCIBILITY STATEMENT

To make our method reproducible, we provide the details of the method in Section 3. We also give the derivations of the evidence lower bound proposed in Section 3 in Appendix A. The datasets used in this paper can be obtained by referencing Section 5. There are also the experimental settings and implementation details for each dataset in Section 5. We also provide an anonymous link for the code of our method, which is in the abstract of this paper.

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

# A   DERIVATION OF THE ELBO FOR MULTI-TASK NEURAL PROCESSES

We provide a derivation of ELBO of the proposed multi-task processed with hierarchical context modeling. The likelihood of multi-task learning is as follows:

$$
\begin{aligned}
p(\{\mathbf{Y}_l^*\}|\{\mathbf{X}_l^*\}, \{D_l\}) &= \prod_{l=1}^{L} p(\mathbf{Y}_l^*|\mathbf{X}_l^*, \{D_l\}) \\
&= \prod_{l=1}^{L} \int \int p(\mathbf{Y}_l^*|\mathbf{X}_l^*, \boldsymbol{\psi}_l) p_{\theta_1}(\boldsymbol{\psi}_l|\boldsymbol{\alpha}_l, \mathbf{M}) p_{\theta_2}(\boldsymbol{\alpha}_l|D_l) d\boldsymbol{\alpha}_l d\boldsymbol{\psi}_l
\end{aligned}
\tag{12}
$$

Based on conditional independence assumption, we introduce the variational joint posterior distribution factorized as (7). By incorporating the variational posteriors in the log likelihood, we can obtain the ELBO as follows:

$$
\begin{aligned}
&\log p(\{\mathbf{Y}_l^*\}|\{\mathbf{X}_l^*\}, \{D_l\}) \\
&= \sum_{l=1}^{L} \log \int \int p(\mathbf{Y}_l^*|\mathbf{X}_l^*, \boldsymbol{\psi}_l) p_{\theta_1}(\boldsymbol{\psi}_l|\boldsymbol{\alpha}_l, \mathbf{M}) p_{\theta_2}(\boldsymbol{\alpha}_l|D_l) d\boldsymbol{\alpha}_l d\boldsymbol{\psi}_l \\
&= \sum_{l=1}^{L} \log \int \Big\{ \int p(\mathbf{Y}_l^*|\mathbf{X}_l^*, \boldsymbol{\psi}_l) p_{\theta_1}(\boldsymbol{\psi}_l|\boldsymbol{\alpha}_l, \mathbf{M}) \frac{q_{\varphi_1}(\boldsymbol{\psi}_l|D_l^*)}{q_{\varphi_1}(\boldsymbol{\psi}_l|D_l^*)} d\boldsymbol{\psi}_l \Big\} p_{\theta_2}(\boldsymbol{\alpha}_l|D_l) \frac{q_{\varphi_2}(\boldsymbol{\alpha}_l|D_l^*)}{q_{\varphi_2}(\boldsymbol{\alpha}_l|D_l^*)} d\boldsymbol{\alpha}_l \\
&\geq \sum_{l=1}^{L} \Big\{ \mathbb{E}_{q_{\varphi_2}(\boldsymbol{\alpha}_l|D_l^*)} \Big\{ \int p(\mathbf{Y}_l^*|\mathbf{X}_l^*, \boldsymbol{\psi}_l) p_{\theta_1}(\boldsymbol{\psi}_l|\boldsymbol{\alpha}_l, \mathbf{M}) \frac{q_{\varphi_1}(\boldsymbol{\psi}_l|D_l^*)}{q_{\varphi_1}(\boldsymbol{\psi}_l|D_l^*)} d\boldsymbol{\psi}_l \Big\} \\
&\qquad - \mathbb{D}_{\mathrm{KL}}[q_{\varphi_2}(\boldsymbol{\alpha}_l|D_l^*)||p_{\theta_2}(\boldsymbol{\alpha}_l|D_l)] \Big\} \\
&\geq \sum_{l=1}^{L} \Big\{ \mathbb{E}_{q_{\varphi_2}(\boldsymbol{\alpha}_l|D_l^*)} \Big\{ \mathbb{E}_{q_{\varphi_1}(\boldsymbol{\psi}_l|D_l^*)}[p(\mathbf{Y}_l^*|\mathbf{X}_l^*, \boldsymbol{\psi}_l)] - \mathbb{D}_{\mathrm{KL}}[q_{\varphi_1}(\boldsymbol{\psi}_l|D_l^*)||p_{\theta_1}(\boldsymbol{\psi}_l|\boldsymbol{\alpha}_l, \mathbf{M})] \Big\} \\
&\qquad - \mathbb{D}_{\mathrm{KL}}[q_{\varphi_2}(\boldsymbol{\alpha}_l|D_l^*)||p_{\theta_2}(\boldsymbol{\alpha}_l|D_l)] \Big\}.
\end{aligned}
\tag{13}
$$

# B   PROOF OF EXCHANGABILITY AND CONSISTENCY

We further provide theoretical proof to show that the proposed multi-task neural processes are valid stochastic processes, which completes the theory of multi-task neural processes. As the statement in Garnelo et al. (2018b): the conditions, including (finite) exchangeability and consistency, are sufficient to define a stochastic process. In our multi-input multi-output setting, we observe $L$ tasks, $\{\mathbf{X}_l^*\}$ where $\mathbf{X}_l^* = \{x_{l,i}^*\}_{i=1}^{n_l^*}$ and $\{\mathbf{Y}_l^*\}$ where $\mathbf{Y}_l^* = \{y_{l,i}^*\}_{i=1}^{n_l^*}$. $x_{l,i}^*$ denotes the $i$-th target samples from task $l$ and $y_{l,i}^*$ is its corresponding target or label. Here are the two propositions to state the exchangeability and consistency of the proposed multi-task neural processes. We model the functional posterior distribution of the stochastic process by approximating the joint predictive distribution over each target set $p(\{\mathbf{Y}_l^*\}|\{\mathbf{X}_l^*\}, \{D_l\})$, which is conditioned on all context samples $\{D_l\}$.

**Proposition 1** *(Exchangability) For finite* $\boldsymbol{n} = \sum_{l=1}^{L} n_l^*$, *if* $\boldsymbol{\pi} = \{\pi_l\}_{l=1}^{L}$ *is a permutation of* $\{1, ..., \boldsymbol{n}\}$ *where* $\pi_l$ *is a permutation of the corresponding order set* $\{1, ..., n_l^*\}$, *then:*

$$
p(\boldsymbol{\pi}(\{\mathbf{Y}_l^*\})|\boldsymbol{\pi}(\{\mathbf{X}_l^*\}), \{D_l\}) = p(\{\mathbf{Y}_l^*\}|\{\mathbf{X}_l^*\}, \{D_l\}),
\tag{14}
$$

*where* $\boldsymbol{\pi}(\{\mathbf{Y}_l^*\}) := (\pi_1(\mathbf{Y}_1^*), ..., \pi_L(\mathbf{Y}_L^*)) = (y_{\boldsymbol{\pi}(1)}^*, ..., y_{\boldsymbol{\pi}(\boldsymbol{n})}^*)$ *and* $\boldsymbol{\pi}(\{\mathbf{X}_l^*\}) := (\pi_1(\mathbf{X}_1^*), ..., \pi_L(\mathbf{X}_L^*)) = (x_{\boldsymbol{\pi}(1)}^*, ..., x_{\boldsymbol{\pi}(\boldsymbol{n})}^*)$.

*Proof.*

$$p(\boldsymbol{\pi}(\{\mathbf{Y}_l^*\})|\boldsymbol{\pi}(\{\mathbf{X}_l^*\}), \{D_l\})$$

$$= \int \int \Big(\prod_{i=1}^{\boldsymbol{n}} p(y_{\boldsymbol{\pi}(i)}^*|x_{\boldsymbol{\pi}(i)}^*, \boldsymbol{\psi}_{1:L})\Big) p(\boldsymbol{\psi}_{1:T}|\boldsymbol{\alpha}_{1:T}, \mathbf{M}) p(\boldsymbol{\alpha}_{1:T}|\{D_l\}) d\boldsymbol{\psi}_{1:L} d\boldsymbol{\alpha}_{1:L}$$

$$= \int \int \Big(\prod_{l=1}^{L}\prod_{j=1}^{n_l^*} p(y_{\pi_1(j)}^*|x_{\pi_1(j)}^*, \boldsymbol{\psi}_l)\Big) \Big(\prod_{l=1}^{L} p(\boldsymbol{\psi}_l|\boldsymbol{\alpha}_l, \mathbf{M})\Big) \Big(\prod_{l=1}^{L} p(\boldsymbol{\alpha}_l|D_l)\Big) d\boldsymbol{\psi}_{1:L} d\boldsymbol{\alpha}_{1:L} \quad (15)$$

$$= \int \int \Big(\prod_{l=1}^{L}\prod_{j=1}^{n_l^*} p(y_j^*|x_j^*, \boldsymbol{\psi}_l)\Big) \Big(\prod_{l=1}^{L} p(\boldsymbol{\psi}_l|\boldsymbol{\alpha}_l, \mathbf{M})\Big) \Big(\prod_{l=1}^{L} p(\boldsymbol{\alpha}_l|D_l)\Big) d\boldsymbol{\psi}_{1:L} d\boldsymbol{\alpha}_{1:L}$$

$$= p(\{\mathbf{Y}_l^*\}|\{\mathbf{X}_l^*\}, \mathbf{M})$$

$\square$

**Proposition 2** *(Consistency) Given* $\boldsymbol{m} = \sum_l^L m_{l=1}$, *if* $1 \leq \boldsymbol{m} \leq \boldsymbol{n}$ *or for each task* $1 \leq m_l \leq n_l^*$, *then:*

$$\int p(\{\mathbf{Y}_l^*\}|\{\mathbf{X}_l^*\}, \{D_l\}) d(\{\mathbf{Y}_l^*\})_{\boldsymbol{m}+1:\boldsymbol{n}} = p((\{\mathbf{Y}_l^*\})_{1:\boldsymbol{m}}|(\{\mathbf{X}_l^*\})_{1:\boldsymbol{m}}, \{D_l\}), \quad (16)$$

*where* $(\{\mathbf{Y}_l^*\})_{1:\boldsymbol{m}} = ((\mathbf{Y}_1^*)_{1:m_1}, ..., (\mathbf{Y}_L^*)_{1:m_L}) = (y_1^*, ..., y_{\boldsymbol{m}}^*)$ *and* $(\{\mathbf{X}_l^*\})_{1:\boldsymbol{m}} = ((\mathbf{X}_1^*)_{1:m_1}, ..., (\mathbf{X}_L^*)_{1:m_L}) = (x_1^*, ..., x_{\boldsymbol{m}}^*)$.

*Proof.*

$$\int p(\{\mathbf{Y}_l^*\}|\{\mathbf{X}_l^*\}, \{D_l\}) d(\{\mathbf{Y}_l^*\})_{\boldsymbol{m}+1:\boldsymbol{n}}$$

$$= \int \int \int \Big(\prod_{i=1}^{\boldsymbol{n}} p(y_i^*|x_i^*, \boldsymbol{\psi}_{1:L})\Big) \Big(\prod_{l=1}^{L} p(\boldsymbol{\psi}_l|\boldsymbol{\alpha}_l, \mathbf{M})\Big) \Big(\prod_{l=1}^{L} p(\boldsymbol{\alpha}_l|D_l)\Big) d\boldsymbol{\psi}_{1:L} d\boldsymbol{\alpha}_{1:L} d(\{\mathbf{Y}_l\})_{\boldsymbol{m}+1:\boldsymbol{n}}$$

$$= \int \int \Big(\prod_{i=1}^{\boldsymbol{m}} p(y_i^*|x_i^*, \boldsymbol{\psi}_{1:L})\Big) \Big(\int \prod_{i=\boldsymbol{m}+1}^{\boldsymbol{n}} p(y_i^*|x_i^*, \boldsymbol{\psi}_{1:L}) d(\{\mathbf{Y}_l^*\})_{\boldsymbol{m}+1:\boldsymbol{n}}\Big)$$

$$\Big(\prod_{l=1}^{L} p(\boldsymbol{\psi}_l|\boldsymbol{\alpha}_l, \mathbf{M})\Big) \Big(\prod_{l=1}^{L} p(\boldsymbol{\alpha}_l|D_l)\Big) d\boldsymbol{\psi}_{1:L} d\boldsymbol{\alpha}_{1:L}$$

$$= \int \int \Big(\prod_{i=1}^{\boldsymbol{m}} p(y_i^*|x_i^*, \boldsymbol{\psi}_{1:L})\Big) \Big(\prod_{l=1}^{L} p(\boldsymbol{\psi}_l|\boldsymbol{\alpha}_l, \mathbf{M})\Big) \Big(\prod_{l=1}^{L} p(\boldsymbol{\alpha}_l|D_l)\Big) d\boldsymbol{\psi}_{1:L} d\boldsymbol{\alpha}_{1:L}$$

$$= p((\{\mathbf{Y}_l^*\})_{1:\boldsymbol{m}}|(\{\mathbf{X}_l^*\})_{1:\boldsymbol{m}}, \mathbf{M})$$

$$(17)$$

$\square$

## C    MORE EXPERIMENTAL DETAILS

Details of iteration numbers and batch sizes for different benchmarks are provided in Table 6. In each batch, the number of training samples from each task and category is identical. We train all models and parameters by the Adam optimizer Kingma & Ba (2014) using an NVIDIA Tesla V100 GPU. The learning rate is initially set as $1e-4$ and decreases with a factor of $0.5$ every $3,000$ iterations. The network architectures of the proposed multi-task neural processes for multi-task classification are provided as follows.

Table 6: The iteration numbers and batch sizes on different datasets, where $C$ and $L$ denotes the number of classes and tasks in the specific dataset, respectively.

| Dataset | Iteration | Batch size |
|---|---|---|
| Office-Home | 15,000 | $8 * C * L$ |
| Office-Caltech | 15,000 | $8 * C * L$ |
| ImageCLEF | 15,000 | $8 * C * L$ |

Table 7: The architecture of inference networks $\varphi_1(\cdot)$ for latent variable $\psi$ of multi-task neural processes.

| Output size | Layers |
|---|---|
| 4096 | Input feature |
| 4096 | Dropout (p=0.7) |
| 4096 | Fully connected, ELU |
| 4096 | Fully connected, ELU |
| 4096 | Reparameterization to $\mu_\psi, \sigma_\psi^2$ |

Table 8: The architecture of inference networks $\varphi_2(\cdot)$ for the latent variable $\alpha$ of multi-task neural processes.

| Output size | Layers |
|---|---|
| 4096 | Input feature |
| 4096 | Dropout (p=0.7) |
| 2048 | Fully connected, ELU |
| 2048 | Fully connected, ELU |
| 2048 | Reparameterization to $\mu_\alpha, \sigma_\alpha^2$ |

Table 9: The architecture of the neural network $h(\cdot)$ of multi-task neural processes.

| Output size | Layers |
|---|---|
| 2048 | Input feature |
| 1024 | Fully connected, ELU |
| 512 | Fully connected, ELU |
| $L$ | Fully connected |
| $L$ | Normalization |
| 4096 | Multiply with the global variable |

The architecture of the inference network $\varphi_1$ is provided in Table 7. The architecture of the inference network $\varphi_2$ is provided in Table 8. We note that the inference network $\theta_1$ and $\theta_2$ share the same architectures with $\varphi_1$ and $\varphi_2$, respectively. The architecture of the neural network $h$ is provided in Table 9. The network is needed because it provides a data-driven way for the model to incorporate the task-specific latent variable $\alpha_l$ and the global variable $M$, which are usually defined in different feature spaces. During inference, we apply the reparameterization trick to generate the samples for the latent variables (Kingma & Welling, 2013).

## D  MORE EXPERIMENTAL RESULTS

### D.1  MULTI-TASK REGRESSION AND CLASSIFICATION WITH THE 20% SPLIT

Further, we provide experiments results on the three multi-task classification datasets with 20% training samples in Table 10, 11 and 12. The proposed multi-task neural processes consistently achieve the best performance on all three benchmarks.

Table 10: Performance comparison of different methods on the `Office-Home` dataset with 20% training samples.

| Methods | A | C | P | R | Avg. |
|---|---|---|---|---|---|
| Single task learning | $54.6_{\pm0.4}$ | $50.6_{\pm0.4}$ | $81.3_{\pm0.2}$ | $73.1_{\pm0.3}$ | $64.9_{\pm0.1}$ |
| Bakker & Heskes (2003) | $61.3_{\pm0.2}$ | $56.5_{\pm0.2}$ | $81.7_{\pm0.3}$ | $75.4_{\pm0.2}$ | $68.7_{\pm0.2}$ |
| Long et al. (2017) | $65.1_{\pm0.3}$ | $46.7_{\pm0.2}$ | $79.9_{\pm0.3}$ | $76.6_{\pm0.3}$ | $67.1_{\pm0.1}$ |
| Kendall et al. (2018) | $59.5_{\pm0.3}$ | $53.8_{\pm0.3}$ | $80.1_{\pm0.1}$ | $73.6_{\pm0.4}$ | $66.8_{\pm0.2}$ |
| Qian et al. (2020) | $58.3_{\pm0.2}$ | $53.5_{\pm0.3}$ | $79.8_{\pm0.2}$ | $73.1_{\pm0.3}$ | $66.2_{\pm0.1}$ |
| Multi-task neural processes | $64.2_{\pm0.1}$ | $55.7_{\pm0.3}$ | $82.6_{\pm0.2}$ | $77.2_{\pm0.3}$ | $\mathbf{69.9_{\pm0.3}}$ |

Table 11: Performance comparison of different methods on the `Office-Caltech` dataset with 20% training samples.

| Methods | A | W | D | C | Avg. |
|---|---|---|---|---|---|
| Single task learning | $94.9_{\pm0.2}$ | $92.8_{\pm0.4}$ | $95.2_{\pm0.6}$ | $86.7_{\pm0.6}$ | $92.4_{\pm0.3}$ |
| Bakker & Heskes (2003) | $95.2_{\pm0.2}$ | $94.4_{\pm0.4}$ | $99.5_{\pm0.3}$ | $91.3_{\pm0.1}$ | $95.1_{\pm0.1}$ |
| Long et al. (2017) | $95.5_{\pm0.3}$ | $94.9_{\pm0.1}$ | $99.2_{\pm0.3}$ | $91.0_{\pm0.4}$ | $95.1_{\pm0.1}$ |
| Kendall et al. (2018) | $95.4_{\pm0.7}$ | $93.2_{\pm0.4}$ | $99.2_{\pm0.4}$ | $91.2_{\pm0.3}$ | $94.7_{\pm0.3}$ |
| Qian et al. (2020) | $95.7_{\pm0.4}$ | $94.1_{\pm0.2}$ | $99.2_{\pm0.5}$ | $91.1_{\pm0.4}$ | $95.0_{\pm0.2}$ |
| Multi-task neural processes | $94.9_{\pm0.3}$ | $96.6_{\pm0.2}$ | $99.2_{\pm0.4}$ | $92.3_{\pm0.2}$ | $\mathbf{95.7_{\pm0.1}}$ |

Table 12: Performance comparison of different methods on the `ImageCLEF` dataset with 20% training samples.

| Methods | C | I | P | B | Avg. |
|---|---|---|---|---|---|
| Single task learning | $92.9_{\pm0.6}$ | $84.6_{\pm0.3}$ | $72.5_{\pm0.4}$ | $54.6_{\pm0.6}$ | $76.2_{\pm0.3}$ |
| Bakker & Heskes (2003) | $94.4_{\pm0.5}$ | $90.6_{\pm0.4}$ | $74.2_{\pm0.4}$ | $57.9_{\pm0.3}$ | $79.3_{\pm0.4}$ |
| Long et al. (2017) | $94.4_{\pm0.4}$ | $89.2_{\pm0.5}$ | $75.8_{\pm0.5}$ | $59.4_{\pm0.3}$ | $79.7_{\pm0.3}$ |
| Kendall et al. (2018) | $93.3_{\pm0.4}$ | $91.0_{\pm0.2}$ | $75.6_{\pm0.2}$ | $56.9_{\pm0.4}$ | $79.2_{\pm0.3}$ |
| Qian et al. (2020) | $93.1_{\pm0.3}$ | $92.1_{\pm0.5}$ | $74.4_{\pm0.7}$ | $55.8_{\pm0.6}$ | $78.9_{\pm0.5}$ |
| Multi-task neural processes | $92.1_{\pm0.4}$ | $91.5_{\pm0.5}$ | $79.2_{\pm0.4}$ | $60.6_{\pm0.3}$ | $\mathbf{80.8_{\pm0.1}}$ |

## D.2 MULTI-TASK REGRESSION WITH LESS DATA

To show the advantages of our model, we compare them on the setting of less data with the 0.05% split. In this case, there are only 20 samples per task during training. As shown in the Table 13, the improvement of our method becomes larger.

Table 13: Performance (Average NMSE) on Rotated MNIST(0.05% split).

| Methods | 0 | 1 | 2 | 3 | 4 | 5 | 6 | 7 | 8 | 9 | Avg. NMSE |
|---|---|---|---|---|---|---|---|---|---|---|---|
| NPs | $0.287_{\pm0.018}$ | $0.149_{\pm0.010}$ | $0.136_{\pm0.007}$ | $0.178_{\pm0.013}$ | $0.135_{\pm0.003}$ | $0.142_{\pm0.007}$ | $0.214_{\pm0.005}$ | $0.198_{\pm0.009}$ | $0.139_{\pm0.006}$ | $0.148_{\pm0.004}$ | $0.173_{\pm0.003}$ |
| MTNPs | $0.185_{\pm0.018}$ | $0.150_{\pm0.013}$ | $0.169_{\pm0.008}$ | $0.132_{\pm0.015}$ | $0.173_{\pm0.004}$ | $0.138_{\pm0.010}$ | $0.196_{\pm0.004}$ | $0.152_{\pm0.010}$ | $0.131_{\pm0.007}$ | $0.165_{\pm0.005}$ | $\mathbf{0.159_{\pm0.002}}$ |

## D.3 STABILITY OF OUR METHOD

The computational advantage of our method can be illustrated by the training loss as function of iteration on *Office-Home*. As shown in Fig. 4, our MTNPs converges more stable than NPs under 5%, 10% and 20% train-test splits. Moreover, we have added a new experiment to investigate the training stability by introducing the noise to the input data.

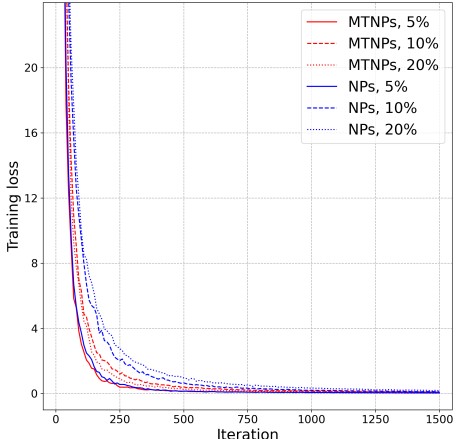

Figure 4: Illustration of training loss with iterations on `Office-Home`. MTNPs converges more stable than NPs under $5\%$, $10\%$ and $20\%$ train-test splits.

We apply the the fast gradient sign method (Goodfellow et al., 2014) to generate the noise. The results are given in Table 14, where $\eta$ denotes the noise level. We observe that our MTNPs show better stability than NPs at different noise levels.

Table 14: The performance under different noise level on the `Office` dataset with $5\%$ split.

| $\eta$ | 0.0 | 0.2 | 0.4 | 0.6 | 0.8 | 1.0 |
|---|---|---|---|---|---|---|
| NPs | 92.6 | 89.5 | 86.6 | 82.7 | 77.7 | 75.1 |
| MTNPs | **94.5** | **91.2** | **87.9** | **84.3** | **82.5** | **81.0** |

## D.4 SENSITIVITY OF THE HYPERPARAMETER $N_f$ AND $N_a$

In practice, we set $N_f$ and $N_a$ to be 10 and 5, which offer a good balance between performance and efficiency. We determine them by grid search as shown in Table 15 and 16.

Table 15: Sensitivity of $N_f$ (and $N_a = 5$) on Office-Home with the 5% split.

| $N_f$ | 1 | 5 | 10 | 20 | 30 |
|---|---|---|---|---|---|
| Avg. | 59.8±0.1 | 59.9±0.1 | **60.0±0.1** | 59.9±0.1 | **60.0±0.0** |

Table 16: Sensitivity of $N_a$ (and $N_f = 10$) on Office-Home with the 5% split.

| $N_a$ | 1 | 5 | 10 | 20 | 30 |
|------|------|------|------|------|------|
| Avg. | $59.8\pm0.1$ | $\mathbf{60.0\pm0.1}$ | $59.6\pm0.1$ | $59.9\pm0.1$ | $59.8\pm0.1$ |

