# OpenReview forum: "Multi-Task Neural Processes"
_ICLR.cc/2022/Conference — ICLR 2022 Submitted_

### Official Review · Reviewer_3xsW · 2021-10-30

**Correctness:** 4
**Technical Novelty And Significance:** 2
**Empirical Novelty And Significance:** 4
**Recommendation:** 6
**Confidence:** 4

**Main Review:**

The strong point is the extensive experiments. The authors have done extensive experiments to evaluate the proposed idea on various tasks. The results are very promising and significant.

The weakness is possibly the weak theatrical contribution. The hierarchical model design and inference are straightforward. The authors did not compare with other possible solutions to show why this design is the best. The design of the original NP is guaranteed to be a valid stochastic process. It is pity that this discussion is missed for multi-task NP. Is multi-task NP still a valid stochastic process? is any marginal NP on a single task in multi-task NP still a valid stochastic process?

**Summary Of The Paper:**

This paper proposes a new multi-task neural process based on the classical neural process. The idea is to introduce additional global variables to share knowledge from different tasks. That is to say, a hierarchical bayesian model is constructed to link single-task neural processes together. The new model is able to handle multi-task applications contract to the classical neural process

**Summary Of The Review:**

The paper is clearly organized and written. It is easy to follow. The results are very impressive. Although some important discussions are missed, this contribution of this work is good.

---

> ### Comment · Reviewer_3xsW · 2021-11-16
> **More details**
>
> The strong points:
>
> 1. it is an extension of the Neural Process family, which is an emerging tool for machine learning, including NP, CNP, Convolutional NP, attentive CNP, etc. This work is another reasonable extension of this family.
> 2. As the authors claimed, this NP-based multi-task learning algorithm is with better ability to handle limited data within each task. This is a benefit of the stochastic process nature of NP. The generalization ability of NP helps it make a better prediction on unseen data. So, compared with other multi-task learning algorithms, the proposed NP-based multi-task learning algorithm is with additional merits. It is a good contribution to this multi-task learning area.
> 3. The uncertainty modeling and prediction is also the nature of NP-based algorithms. The proposed learning algorithm is a new tool for uncertainty modeling of multi-task learning.
> 3. the experiments are extensive. Different tasks, like classification and regression, brain segmentation, are conducted to show the comparative advantages of the proposed idea.
>
> The weak points：
>
> 1.  the most important one is the stochastic process nature of the proposed multi-task NP. In other NP family members, the latent variable z is aggregated using a permutation invariant operation to ensure the stochastic process nature, including exchangeability and consistency requirements. It is important because it is the source of the generalization ability for limited data. However, the authors did not discuss that.
> 2. the variable M is with shadow in Figure 1(b), so it is apparently an observed variable, but was not clearly explained. I guess it is just a collection of samples from different classes but not sure. How to ensure it is with shared and transferable knowledge of different tasks?
> 3. the whole model is simply constructed and the amortized VI is also straightforward. There is not much difficulty with that.
> 4. since the strong point of the proposed algorithm is to handle limited data in each task. The experiments should be designed for this setting. The various multi-task GP should be compared because they can also handle limited data using stochastic process nature. The uncertainty should be output and compared with multi-task GP.

---

> > ### Author Response · Authors · 2021-11-23
> > **Response to Reviewer 3xsW**
> >
> > **Main comments**
> >
> > **Q1**: *Theoretical contributions*
> >
> > **A1**: We summarize the theoretical contributions of this paper as follows. We have added this in Sections 1 & 3.
> > * Our multi-task neural processes are a methodological extension of neural processes for multi-task learning, which fills the theoretical gap of neural processes for multi-task learning.
> > *  In this revised version, we further provide theoretical proof to show that the proposed multi-task neural processes are valid stochastic processes, which completes the theory of multi-task neural processes. This detailed proof is given in Appendix B.
> > * The designed hierarchical context modeling provides a principled way to explore task relatedness in the function space, which allows task-specific function variables to leverage the shared knowledge from related tasks.
> >
> > **Q2**: *Possible solusions*
> >
> > **A2**: The neural processes with all task context are an alternative solution, which treats context sets from all tasks equally without hierarchical context modeling. We conducted comparison experiments. The results in Table 1 show that the proposed design performs better. The hierarchical context modeling can better explore the task relatedness in function space, which enables the models to capture the relevant knowledge from related tasks even in presence of distribution shift among tasks. We have added this discussion in Section 5.2.
> >
> > **Q3**: *Valid stochastic processes*
> >
> > **A3**: The proposed multi-task neural processes are valid stochastic processes since they satisfy the conditions of the Kolmogorov Extension Theorem.  This detailed proof is given in Appendix B. The marginal neural process for each task is also a valid stochastic process since based on the conditional independence it degrades to a regular neural process for a single task.
> >
> >
> > **Details**
> >
> > **Q1**: *Stochastic process nature*
> >
> > **A1**: The proposed multi-task neural processes also enjoy the properties of exchangeability and consistency. In this revised version, we have provided rigorous theoretical proof associated with discussions, which have been added in Appendix B.
> >
> > **Q2**: *The global variable $\mathbf{M}$*
> >
> > **A2**:
> > We apologize for not making the variable $\mathbf{M}$ sufficiently clear.
> > The reviewer is correct and $\mathbf{M}$ collects samples from all $L$ tasks.
> >
> > For regression tasks, $\mathbf{M} \in \mathbb{R}^{L*d}$ and each row is a feature vector which is the average of all feature vectors from a task.
> >
> > For classification tasks, $\mathbf{M} \in \mathbb{R}^{{L*C}*d}$ where each vector $\mathbf{M}_{l,c}$ is the average of all features of each category from a task.
> >
> > The shared and transferable knowledge is ensured because tasks share the same target space. For regression, tasks share the same rotation angle space. For classification, tasks share the same category space.
> > We have added the explanations in Section 3.1.
> >
> > **Q3**: *Training with limited data & predicting uncertainty*
> >
> > **A3**: Thank you for sharing the insight.
> >
> > * We conduct the experiments under the setting of limited training data. For multi-task classification datasets, we use 5\%, 10\% and 20\% labeled data for training, which correspond to about 3, 6 and 12 samples per category per task, respectively.
> >
> > * We have added the experiment to compare with multi-task Gaussian processes. The experiment is conducted on the multi-task regression dataset using the Rotated MNIST dataset.
> > We use the average normalized variance to measure the uncertainty of the prediction.
> > As shown in the table below, our method has a lower average NMSE and average normalized variance. We have added this in Section 5.3. Thank you.
> >
> > | Methods                         | Average NMSE    | Average normalized variance |
> > |---------------------------------|-----------------|-----------------------------|
> > | Multi-Task Gaussian process [a] | 0.153±0.000     | 0.017                       |
> > | MTNPs                           | **0.106±0.001** | **0.003**                   |
> >
> > Reference:
> >
> > [a] Yu et. al., Learning Gaussian processes from multiple tasks, ICML2005.

---

> > > ### Comment · Reviewer_3xsW · 2021-11-29
> > > **Acknowledgement of Rebuttal**
> > >
> > > Thanks for the author's responses!
> > >
> > > The authors have clarified some concerns, like stochastic nature, M, and some experiments.
> > >
> > > I still have the following comments for this work:
> > >
> > > 1. The selected multi-task GPs paper is old. There are some latest ones that are more appropriate to be compared in this work, like
> > > "Deep Multi-task Gaussian Processes for Survival Analysis with Competing Risks", NIPS2017.
> > >
> > > 2. I agree with the authors that this work is a "methodological extension" of NP but I did not see any "theoretical gap of neural processes for multi-task learning". What do you mean by this 'theoretical gap'? In my view, this work is just a "methodological extension" of NP for the multi-task problems with a straightforward hierarchical Bayesian method. Yes, this work is a new thing to the multi-task area but not with much theoretical contribution.

---

> > > > ### Author Response · Authors · 2021-11-30
> > > > **Further response to Reviewer 3xsW**
> > > >
> > > > We thank you for your response, engagement and encouragement.
> > > >
> > > > **A1:**
> > > > We choose Yu et. al, ICML2005 as our baseline because they also adopt a hierarchical architecture to model task relatedness, which can provide a head-to-head comparison with our model.
> > > > We have looked into the paper (Alaa \& van der Schaar, NIPS2017).
> > > > Their method assumes all task-specific outputs are statistically dependent. However, it is not the case in our setting, which makes it nontrivial to directly compare with this method. We will definitely add these discussions. Thank you for bringing this interesting reference.
> > > >
> > > > **A2:**
> > > > Thank you for your agreement on our methodological extension of neural processes.
> > > > The theoretical gap here is whether there exists a theoretical guarantee for multi-task neural processes being valid stochastic processes. We provide the theoretical proof to show that our multi-task neural processes are valid stochastic processes.
> > > > The proof indicates that without the hierarchical context modeling, mathematically the multi-task neural processes degrade to standard neural processes.
> > > > While the theoretical contribution of this work may appear modest, its conceptual and practical impact on multi-task learning is considerable.
> > > > We will add these discussions in the paper.
> > > >
> > > > We hope to have mended the remaining concerns and thank you for the help to improve the paper.

---

### Official Review · Reviewer_Ef49 · 2021-11-03

**Correctness:** 2
**Technical Novelty And Significance:** 2
**Empirical Novelty And Significance:** 1
**Recommendation:** 5
**Confidence:** 4

**Main Review:**

Strength:
Even though rather limited, the proposed method shows better performance over compared methods for most of the learning tasks in their experiment.

Weakness:
The paper is poorly written, many places lack of clarity. (1) The description of the method is very confusing. In the main text, on page 4, it seems to suggest the \mu of and \Sigma of \psi are functions of m_l. However, in the network specifications in appendix, it seems \mu of and \Sigma of \psi has nothing to with m_l, because the input size of the \psi_1 network is 4096 but output size of h network is L. In addition, it is not clear why the h network is even needed. I did not see it is involved in either Eq. (12) or (13) to make predictions. (2) In section 3.1 it is mentioned that concrete formation of global variable M depends on the learning scenarios. However, it is not made clear anywhere in the paper how M is constructed from the context data. (3) It is unclear the multi-task setup in the brain image segmentation experiment. (4) How many Monte Carlo samples, i.e., N_f and N_a are drawn for the variational posteriors \psi and \alpha? And how they were determined?

Are networks \psi_1 and \psi_2 common networks shared among all tasks? Or are they task specific? If both are task specific, I do not see mathematically it would make much difference to train one single network comparing to the two that the authors chose to do. In other words, I do not see a clear value of proposed hierarchical design mathematically. More discussion on this would be very helpful.

More empirical studies to support their claims are desired from both aspects of the methods to compare and learning tasks to test on. Most of the recently published multi-task learning methods were neither discussed or compared, for example Quo et. al., 2020, ICML, Sun et. al., 2020, NeurIPS, and Yu et. al., 2020 NeurIPS. The learning tasks used are limited to image recognition related tasks. The results could be more persuasive if tasks from diverse domains are included.


**Summary Of The Paper:**

This paper presents a multi-task neural processes approach in which function priors are derived in a hierarchical Bayesian inference framework to incorporate the shared knowledge into its context of the prediction function. Authors introduced a higher-level latent variable derived from the context data of related tasks to control the sharing of common knowledge between the tasks. Previous works used context data from its own task to generate the task-specific latent variables, whereas this work uses data from other tasks as well. The shared knowledge from all the tasks added to context of each individual task acts as inductive bias for predictions. Their experiment results show better performance of the proposed method than compared ones.

**Summary Of The Review:**

The paper is poorly written. The motivation of the proposed formulation is not clear. The empirical study is not adequate.

---

> ### Author Response · Authors · 2021-11-23
> **Response to Reviewer Ef49**
>
> **Q1:**  *Lack of clarity*
>
> **A1:** (1) *h network*
>
> We regret that our descriptions were not clear to you.
> The reviewer is right. We have corrected the mistake in Table 9 of Appendix C. The output size of h network should be 4096.
> The network is needed because it provides a data-driven way for the model to integrate the task-specific latent variable $\alpha_l$ and the global variable $M$ that are in different feature spaces. During the inference, the network is used to generate the prior distribution of the latent variable $\psi_l$ in Eq. (10) or (11) of the revised paper. We have further clarified this in Section 3.2.
>
> (2) *Model construction*
>
> We apologize for not describing the global variable M clearly. We have added the descriptions in Section 3.1.
>
> For regression tasks, $\mathbf{M} \in \mathbb{R}^{L*d}$ and each row vector corresponds to one task, which is the average of all features from the corresponding task.
>
> For classification tasks, $\mathbf{M} \in \mathbb{R}^{{L*C}*d}$ where each vector $\mathbf{M}_{l,c}$ is the average of all features of each category from the corresponding task.
>
> (3) *Setup*
>
> We reformulate the segmentation task as a pixel-wise regression problem, where each pixel corresponds to a regression task to predict the probability of this pixel belonging to the tumor. In doing so, the spatial correlation and dependency among pixels are effectively modeled by capturing the task relatedness. We have added the details in Section 5.4.
>
> (4) *Monte Carlo samples*
>
> In practice, we set $N_f$ and $N_a$ to be 10 and 5, which offers a good balance between performance and efficiency. They are determined by grid search as shown in the following tables. We have added these results in Appendix D.4.
>
>  * Sensitivity of $N_f$ (when $N_a$ = 5) on Office-Home with the 5% split.
>
> | $N_f$ | 1| 5 | 10| 20| 30 |
> |-------|----------|----------|--------------|----------|--------------|
> | Avg.  | 59.8±0.1 | 59.9±0.1 | **60.0±0.1** | 59.9±0.1 | **60.0±0.0** |
> |
> * Sensitivity of $N_a$ (and $N_f=10$) on Office-Home with the 5\% split.
>
> | $N_a$ | 1| 5 | 10| 20| 30 |
> |-------|----------|--------------|----------|----------|----------|
> | Avg.  | 59.8±0.1 | **60.0±0.1** | 59.6±0.1 | 59.9±0.1 | 59.8±0.1 |
> |
>
> **Q2:** *Inference networks and hierarchical design*
>
> **A2:** Thank you for your suggestion. Both networks $\varphi_1$ and $\varphi_2$ are amortized inference networks and shared by all tasks. Mathematically, the value of the proposed hierarchical design is that it enables the model to capture the relevant knowledge from the global variable in presence of distribution shift among tasks.
> In particular, the prior takes the form $p(\psi_l|M) = \int p(\psi_l|\alpha_l, M)p(\alpha_l|D_l)d\alpha_l$, where the global knowledge $M$ is further adaptively processed by the hyper-level latent variable $\alpha_l$ to generate task-specific classifiers/regressors $\psi_l$.
> We have added the discussions in Section 3.1.
>
> **Q3:** *Compared methods and tasks from diverse domains*
>
> **A3:** We thank the reviewer for bringing these interesting references, which we have included in the related work of the revised paper for discussions.
> * We have added experiments to compare with Guo et. al., 2020, ICML as a representative work that performs better than others. The experiments are conducted on four benchmarks for both multi-task regression and multi-task classification. As shown in the following table, the proposed multi-task neural processes consistently perform better. The reason would be that Guo et. al., 2020, ICML relies on large amounts of training data and therefore tends to overfit under our setting with limited data. We have added these experiments and discussions in Section 5.
>
> | Datasets  | Rotated MNIST   |Office-Home   |  |  | Office-Caltech |  |   | ImageCLEF    |  |  |
> | -- | -- | -- | -- | -- | -- | -- | -- | -- | -- | -- |
> | Split  | 0.1\%| 5\%| 10\%| 20\%| 5\% | 10\% | 20\%| 5\%| 10\% | 20\%|
> | Guo et. al., 2020, ICML | 0.109±0.002    | 38.3±0.5    | 51.5±0.3    | 62.2±0.4    | 74.6±0.4      | 80.4±1.2    | 89.9±0.8    | 51.7±0.9    | 62.6±0.8   | 71.6±0.4    |
> | MTNPs | **0.106±0.001** | **60.0±0.1** | **63.3±0.1** | **69.9±0.3** | **94.6±0.1**  | **95.4±0.1** | **95.7±0.1** | **76.1±0.1** | **79.6±0.1** | **80.8±0.1** |
> * To include more diverse domains, we have tested the proposed multi-task neural processes on the 1-D synthetic example. The results show that the predictions of the proposed multi-task neural processes more resemble the ground truth than that of neural processes, especially in the boundary of different tasks. We have provided the experimental details and results in Section 5.1 and Figure 2. Thank you.
>
> References:
>
> [a] Guo et. al., Learning to Branch for Multi-Task Learning, ICML2020.
>
> [b] Sun et. al., AdaShare: Learning What To Share For Efficient Deep Multi-Task Learning, NeurIPS2020.
>
> [c] Yu et. al., Gradient Surgery for Multi-Task Learning, NeurIPS2020.

---

### Official Review · Reviewer_H8nA · 2021-11-05

**Correctness:** 3
**Technical Novelty And Significance:** 2
**Empirical Novelty And Significance:** 2
**Recommendation:** 5
**Confidence:** 4

**Main Review:**

Many Bayesian models have been proposed for multi-task learning. Why does the proposed multi-task neural process achieve superior performance? It is better to compare different models.

The Gaussian likelihood is not suitable for classification tasks. Authors should consider other likelihood function.

In experiments, important baselines are missing. Authors should compare with MTL models such as cross-stitch network and MTAN, which can be modified to solve the multi-input multi-output setting. Moreover, multi-task Gaussian process should be compared.

**Summary Of The Paper:**

This paper proposes a multi-task neural process as a multi-task Bayesian model.

**Summary Of The Review:**

This paper proposes a multi-task Bayesian model. Compared with existing multi-task Bayesian models, I don't know why the proposed model is better. The proposed model is not suitable for classification tasks. Important baselines are missing in experiments.

---

> ### Author Response · Authors · 2021-11-23
> **Response to Reviewer H8nA**
>
> **Q1:** *Superior performance*
>
> **A1:** Following your suggestion, we have added comparisons with several Bayesian models, including Variational Single-Task Learning, Variational Multi-Task Learning, Neural Processes and the Bayesian Multi-Task Learning by Bakker \& Heskes (2003). Experimental results show that the proposed multi-task neural processes produce superior performance. This is because our multi-task neural processes directly infer the parameters of prediction functions rather than the input representation, which is able to model a broader range of functional distribution. Moreover, in function space, the hierarchical context modeling can better explore the task relatedness, which enables the models to capture the relevant knowledge even in presence of distribution shift among tasks. We have added the discussions in Section 5.2.
>
> **Q2:** *Gaussian likelihood*
>
> **A2:** The reviewer is correct. We use the Gaussian likelihood for regression tasks. For classification tasks, we use $\sum^{n}_{i=1}{y_i\log(f(x_i))}$ as the log-likelihood function. We have added this in Section 3.1. Thank you.
>
> **Q3:** *Important baselines*
>
> **A3:** Thank you for your suggestion. We have implemented the suggested methods including multi-task Gaussian process [a] and MTAN [b] and compared our method with them. The experiments are conducted for multi-task regression on Rotated MNIST. As shown in the following table, our method performs better. We have added this in Table 5 of Section 5.3. Thank you.
>
> | Methods                         | Avg. NMSE       |
> |---------------------------------|-----------------|
> | Multi-task Gaussian process [a] | 0.153±0.000     |
> | MTAN [b]                        | 0.152±0.018     |
> | MTNPs                           | **0.106±0.001** |
>
> References:
>
> [a] Yu et. al., Learning Gaussian processes from multiple tasks, ICML2005.
>
> [b] Liu et. al., End-to-end multi-task learning with attention, CVPR2019.

---

### Official Review · Reviewer_fDzq · 2021-11-06

**Correctness:** 3
**Technical Novelty And Significance:** 3
**Empirical Novelty And Significance:** 3
**Recommendation:** 6
**Confidence:** 3

**Main Review:**

**Strengths:**

The general idea of the model seems to be a valid one for multitask learning, and to my knowledge, it seems original.

For the most part, the experimental section shows the strength of this model and the performance gains it brings against some other models in the literature. I particularly liked the last example on Brain Image segmentation.

**Weaknesses:**

I was a bit confused with the model construction. What exactly is the global variable $\mathbf{M}$? Is it simply the collection of the context datasets for all the tasks? Or is it a latent variable? If it is a latent variable, why do you opt for doing MAP estimation instead of placing a prior over it? I think this needs to be clarified further in the manuscript.

There are also some errors in the model construction section. The Gaussian likelihood is used to derive the generic multitask model. While this is a reasonable choice for regression tasks, it is a wrong assumption for classification tasks which are also considered in this paper. The inference section seems to have fixed this problem though. I suggest that section 3 is revisited to correct those discrepancies.

I would have liked to see a synthetic example exploring the properties of this model. The experiments jump straight to comparison on real-world datasets, but it would have been better to first start with a synthetic dataset to validate the modelling choices in Section (3).

Table (5) is missing confidence intervals. It also looks like the performance difference might not be significantly different from NPs for this example. I think this should be discussed.

Question: do the nested expectations in eq (11) cause any problems with the variance? How stable is the training of this model in comparison to standard neural processes?


**Summary Of The Paper:**

The paper proposes a multi-task learning model with neural processes. The model is based on a hierarchical construction whereby each task is conditioned on global and local information. The paper derives a “hierarchical” ELBO for this model that is evaluated with MC sampling. Experiments are presented to validate the proposed model.

**Summary Of The Review:**

In my opinion, this work is original and has the potential to be an impactful contribution. However, in its current state, it is not ready yet. My main issues are the ambiguity in the model construction exposition and the lack of experiments that explore the model’s behaviour and properties.

**********
Edit: My score was updated to 6 after the rebuttal.
**********

---

> ### Author Response · Authors · 2021-11-23
> **Response to Reviewer fDzq**
>
> **Q1:**  *Model construction*
>
> **A1:**  We apologize for the confusion. The reviewer is correct. The global variable $\mathbf{M}$ collects the context data for all tasks and it is not a latent variable.
>
> For regression tasks, $\mathbf{M} \in \mathbb{R}^{L*d}$ and each row corresponds to one task, which is the average of all feature vectors from a task.
>
> For classification tasks, $\mathbf{M} \in \mathbb{R}^{{L*C}*d}$ where each vector $\mathbf{M}_{l,c}$ is the average of all features of each category from the corresponding task.
> We have clarified this in Section 3.1.
>
> **Q2:**  *Gaussian likelihood*
>
> **A2:**  The reviewer is correct that the Gaussian likelihood is used for regression tasks. For classification tasks, we actually use $\sum^{{n}}_{i=1} {y_i \log(f(x_i))}$ as the log-likelihood. We have added this in Section 3 to correct the discrepancies. Thank you for this suggestion.
>
> **Q3:**  *Synthetic example*
>
> **A3:**  Following your suggestion, we have added experiments on synthetic examples to explore the properties of our model. The detailed experimental settings and the results have been added in Section 5.1 and Figure 2. In the experiments we compare our multi-task neural processes with regular neural processes. The results show that the predictions of the proposed multi-task neural processes more resemble the ground truth than that of neural processes, especially in the boundary of different tasks.
>
> **Q4:** *Confidence intervals*
>
> **A4:**  We have added the 95\% confidence intervals from 5 runs in Table 5.
> The difference is slight due to that models suffer less from over-fitting.
> To show the advantages of our model, we compare them on the setting of less data with the 0.05% split. In this case, there are only 20 samples per task during training. As shown in the table below, the improvement of our method becomes larger.
> We have added the experiments and discussions in Appendix D.2.
>
> | Methods | 0           | 1           | 2           | 3           | 4           | 5           | 6           | 7           | 8           | 9           | Avg. NMSE       |
> |---------|-------------|-------------|-------------|-------------|-------------|-------------|-------------|-------------|-------------|-------------|-----------------|
> | NPs     | 0.287±0.018 | 0.149±0.010 | 0.136±0.007 | 0.178±0.013 | 0.135±0.003 | 0.142±0.007 | 0.214±0.005 | 0.198±0.009 | 0.139±0.008 | 0.148±0.004 | **0.173±0.003** |
> | MTNPs   | 0.185±0.018 | 0.150±0.013 | 0.169±0.008 | 0.132±0.015 | 0.173±0.004 | 0.138±0.010 | 0.196±0.004 | 0.152±0.010 | 0.131±0.007 | 0.165±0.005 | **0.159±0.002** |
>
> **Q5:** *Nested Expectations*
>
> **A5:** Thank you for sharing this insight.
>
> -In our formulation, the nested expectations in eq (11) do not cause any problem with the variance, because the log-likelihood solely depends on the variational posterior $\psi_l$. Thus, the nested expectations can be reduced to one expectation. In practice, we apply the local reparameterization trick [a] to reduce the variance of stochastic gradients.
>
> -For your interest, we have added the training loss curves in Figure 4 of Appendix D.3. The results show that our method converges more stable than regular neural processes. Further, we have added a new experiment to investigate the training stability by introducing the noise to the input data. The experiment is conducted on the Office-Caltech dataset with the 5% split.
> We apply the fast gradient sign method [b] to generate the noise. The results are given in the following table, where $\eta$ denotes the noise level. We observe that our MTNPs show better stability than NPs at different noise levels. We have added this experiment in Table 14 of Appendix D.3. Thank you.
>
> | $\eta$ | 0.0          | 0.2          | 0.4          | 0.6          | 0.8          | 1.0          |
> |--------|--------------|--------------|--------------|--------------|--------------|--------------|
> | NPs    | 92.6±0.0     | 89.8±0.1     | 86.5±0.4     | 82.5±0.3     | 78.1±0.5     | 75.1±0.6     |
> | MTNPs      | **94.5±0.0** | **91.5±0.2** | **87.6±0.3** | **84.4±0.5** | **81.7±0.8** | **80.4±0.5** |
>
>
> References:
>
> [a] Kingma et., al, Variational Dropout and the Local Reparameterization Trick, NeurIPS2015.
>
> [b] Goodfellow et., al, Explaining and harnessing adversarial examples, 2014.

---

> > ### Comment · Reviewer_fDzq · 2021-11-29
> > **Acknowledgement of Rebuttal**
> >
> > Dear authors,
> >
> > Thank you for your response and for taking the effort to address my concerns. I have read your response and you have managed to address more of my concerns. I think the synthetic example you presented showcases some of the strength of this method. In the next revision, I would urge you to explore more of its properties in comparison to the standard NP, in particular, its behaviour near task boundaries (it seems to interpolate those well) and its quantification of uncertainty (it seems to slightly under-estimate uncertainty compared to NPs, see task 1 with 12 context points).
> >
> > I have updated my score to 6 to reflect my satisfaction with your rebuttal.
> >
> > Reviewer fDzq

---

### Author Response · Authors · 2021-11-23
**Summary**

**We thank all reviewers for their insightful reviews, sharp comments and supportive suggestions. Here, we provide a summary of the updates made in the new version, as suggested by the reviewers.**

Main manuscript
-------------------

The following updates have been incorporated into the main manuscript:
* We have clarified theoretical contributions in Sections 1 and 3.
* We have added detailed descriptions of the global variable in Section 3.1.
* We have added the log-likelihood function for classification tasks in Section 3.1.
* We have expanded related works by adding discussions about all suggested references in Section 4.
* We have conducted experiments on the synthetic example and added related discussions in Figure 2 and Section 5.1.
* We have provided discussions about why our method achieves superior performance in Section 5.2.
* We have added the 95\% confidence intervals from 5 runs in Table 5 of Section 5.3.
* We have implemented three suggested methods and added the results in Table 5 of Section 5.3.
* We have clarified the multi-task setup for the brain segmentation dataset in Section 5.4.

Appendix
-------------------

The following updates have been inserted in the Appendix.
* We further provide theoretical proof to show that the proposed multi-task neural processes are a valid stochastic processes, which completes the theory of multi-task neural processes. The proof has been added in the Appendix B.
*  We have corrected the mistake in Table 9.
*  We have added experiments on Rotated MNIST with less data in Table 13 and Appendix D.2.
*  We have added training loss curves in Figure 4 and conducted the experiment which investigates how stable the training is in Table 14 and Appendix D.3.
* We have added the experiments to show the sensitivity of the hyperparameter $N_f$ and $N_a$ of the Monte Carlo sampling in Table 15 &16 and Appendix D.4.

---

### Decision · Program_Chairs · 2022-01-20

**Decision:**

Reject

**Comment:**

This paper extends neural processes (NPs) to the multi-task setting (MTNPs). The approach uses a hierarchical Bayesian construction, where the latent variables of an NP are conditioned on a set of global task-specific context variables. This allows the NP to share knowledge across related tasks.

There were a few issues raised in the reviews. Consistently, the reviewers noted that the writing could be improved. There were variables, like the context variables M, that lacked explanation. There was also confusion between the use of a Gaussian likelihood for classification vs regression. These were resolved with the author’s response and updated draft.

There were also requests for additional experiments and baselines: 1) a synthetic task, to which the authors included a 1D regression task. 2) More baselines against other multi-task methods, to which the authors included a comparison to Guo et al., 2020, MTAN, and multi-task Gaussian Processes.

Finally, there were questions around whether MTNPs are valid stochastic processes. This has been proven theoretically by the authors, albeit in the appendix.

Currently, this paper remains borderline. The main remaining criticisms are a) A desire for more experiments and analysis to highlight the particular strengths of the approach. b) That the approach is a straightforward extension of NPs, and may not be sufficiently novel. c) That the authors include more baselines from the recent multi-task literature (Yu et al. and Sun et al.). In the end it was determined that the paper does not quite meet the bar for acceptance. I think in future submissions, it would be worthwhile to further highlight MTNP’s performance in the low-data regime, where it particularly seems to do well, and to complete the full set of comparisons (e.g., Sun et al. and Yu et al.) that were requested by the reviewers.